# High-density cortical µECoG arrays concurrently track spreading depolarizations and long-term evolution of stroke in awake rats

Kay Palopoli-Trojani [1,5], Michael Trumpis [1,5], Chia-Han Chiang[1], Charles Wang[1], Ashley J. Williams[1], Cody L. Evans[2], Dennis A. Turner [1,3,4], Jonathan Viventi [1]✉ & Ulrike Hoffmann [2]✉

Spreading depolarizations (SDs) are widely recognized as a major contributor to the progression of tissue damage from ischemic stroke even if blood flow can be restored. They are characterized by negative intracortical waveforms of up to -20 mV, propagation velocities of 3 - 6 mm/min, and massive disturbance of membrane ion homeostasis. High-density, micro-electrocorticographic (µECoG) epidural electrodes and custom, DC-coupled, multiplexed amplifiers, were used to continuously characterize and monitor SD and µECoG cortical signal evolution in awake, moving rats over days. This highly innovative approach can define these events over a large brain surface area (~ 3.4 × 3.4 mm), extending across the boundaries of the stroke, and offers sufficient electrode density (60 contacts total per array for a density of 5.7 electrodes / mm$^2$) to measure and determine the origin of SDs in relation to the infarct boundaries. In addition, spontaneous ECoG activity can simultaneously be detected to further define cortical infarct regions. This technology allows us to understand dynamic stroke evolution and provides immediate cortical functional activity over days. Further translational development of this approach may facilitate improved treatment options for acute stroke patients.

Acute, thrombotic stroke following large cerebral vessel occlusion remains one of the leading causes of death and long-term disability worldwide[1]. The pathophysiology of thrombotic stroke encompasses an ischemic core of irreversibly damaged brain tissue intermixed with marginal penumbra regions with reduced blood flow[2]. Importantly, the penumbra, although metabolically compromised, is potentially salvageable if the region could both be protected from secondary damage and accrue sufficient collateral blood flow to maintain metabolism. Neuroprotection of penumbral tissue is thus critical to prevent expansion and stabilization of the ischemic core region.

However, the ischemic core and particularly the dynamic penumbra are difficult to delineate over the first several hours (and even more difficult over multiple days) after stroke through any conventional criteria, including blood flow, imaging (clinical MRI, CT, direct optical imaging), and angiography. Due to the complexity of cortical vascular domains and collateral flow between these microdomains, these two adjacent regions may initially

be interlocking patches on the cortical surface and deeper regions below the surface, rather than necessarily a solid or continuous core with an annulus, as often schematically visualized. As the regions demarcate over time (i.e., when the complex dynamics of the penumbra subside and the final infarct volume begins to emerge), residual neurological function, histology, and clinical imaging can identify the merged areas of ischemic damage, particularly after 10–14 days. However, this is typically too late to save marginal regions within the penumbra. The fate of the penumbra and overall infarct size are mainly determined by the overall balance of positive factors, such as reperfusion of the primary occluded vessel, reduced metabolic demand, and enhanced collateral blood flow from adjacent vascular distributions, versus negative factors, particularly metabolic insufficiency and critically the occurrence of spreading depolarizations causing high metabolic demand (SDs)[3–5].

Spreading depolarizations (SDs) are waves of neuronal and glial depolarization associated with massive K$^+$ and glutamate efflux, and Ca$^{2+}$

[1]Biomedical Engineering, Duke University, Durham, NC, USA. [2]Center for Perioperative Organ Protection, Department of Anesthesiology, Duke University, Durham, USA. [3]Neurosurgery, Neurobiology, Duke University, Durham, USA. [4]Research and Surgery Services, Durham VAMC, Durham, USA. [5]These authors contributed equally: Kay Palopoli-Trojani, Michael Trumpis. ✉e-mail: j.viventi@duke.edu; Ulrike.Hoffmann@UTSouthwestern.edu

influx, slowly propagating across brain tissue at a rate of 3–6 mm/min. These large, slow cortical events occur after cerebral ischemia in all species studied to date[6,7], including humans[8–10], and are a major contributor to the progression of tissue damage after stroke[11]. With each SD event, the background, spontaneous cortical activity (ECoG) is suppressed due to profound cellular depolarization and inactivation. The cellular energy requirements to recover from this severe depolarization are very high, with a significant demand for increased blood flow to provide substrate (i.e., glucose and oxygen). If this heavy demand cannot be met (due to inadequate collateral) then a terminal depolarization reflects ongoing tissue damage, an indicator of functional deterioration and infarct growth[7,12]. Spreading depolarizations often appear in clusters, and as their frequency increases, the time to terminal depolarization decreases[3].

Preclinical laboratory research studies have focused on the early contribution of SDs to infarct development (i.e., over 4–6 h), with a classical neurophysiology setup based on intracortical microelectrodes (i.e., hollow glass pipettes filled with ionic media and referenced to a distal AG-AgCl ground) and/or imaging through cranial windows to detect extracellular fluid shifts associated with spreading depolarizations. However, these preclinical studies require the humane use of anesthesia, limiting the studies to short-term, non-survival experiments. Since this approach has not allowed the detailed monitoring of SD occurrence over prolonged periods of time and without the confounding effects of anesthesia (i.e., in the awake condition), our understanding of how SDs impact cerebral stroke outcome over time is therefore significantly limited and potentially distorted. Furthermore, this conventional preclinical method of detecting and monitoring SDs is limited to discrete recording sites[13] and does not detect more global brain activity i.e., ECoG across large cortical regions, nor is applicable to the monitoring of stroke patients.

In patients, an invasive monitoring approach involves the implantation of a small linear strip of 4 mm diameter metal ECoG contacts (i.e., 4–6) into the subdural space to detect SDs. The insights learned from monitoring head injury and malignant stroke patients are invaluable but also inevitably skewed (towards very sick patients), as these strips are mainly implanted over areas of severely injured brain[14,15] and only if a craniotomy is necessary to alleviate the increasing intracranial pressure. This invasive approach is typically not feasible in patients with less severe or very focal strokes due to the lack of a clinical rationale for surgery[8]. Since the coverage of the brain is limited by both the small dimensions of the low channel-count strips and the site of the clinically dictate craniotomy, there is often poor detection of SDs unless the strip is directly placed over an involved area. Still, these strips require at least an invasive burr hole or a craniotomy for placement, typically not indicated for most stroke patients. Therefore, we rarely have clinical data on SD characteristics and their contribution to secondary injury after most clinical strokes.

In the pre-clinical setting, chronic, epidural recordings using metal screws implanted into the skull of animals have demonstrated that SDs in awake, non-anesthetized animals may occur at much higher frequency and for longer time intervals than initially believed[13]. However, there are few such chronic studies. Epidural recordings with skull screws lacks the high resolution needed for detailed monitoring of SD characteristics across the brain, and most importantly, the concurrent measurement of brain activity (ECoG) for monitoring of both SDs and high-resolution spontaneous ECoG, which can facilitate the detection of highly dynamic, inhomogeneous stroke boundaries. Thus, a combination of higher-resolution recordings directly from the brain surface with prolonged recordings in awake individuals is urgently needed.

High-density μECoG metal electrodes and multiplexed amplifiers were developed for DC-coupled cortical recordings, which were used to continuously characterize and monitor SD and spontaneous μECoG evolution in awake, moving rats over days. This highly innovative approach can define cortical events across a large brain surface area (~3.4 × 3.4 mm), extending across the boundaries of stroke regions, and also offers sufficient electrode density (60 contacts total per array for a density of 5.7 electrodes/mm²) to measure and determine the origin of SDs in relation to infarct boundaries.

This technology allows us to understand dynamic stroke evolution and provides immediate diagnostic feedback of cortical function over days. Our high-density, 60-channel electrode arrays can reliably assess the pattern of spread and frequency of SD occurrence as well as the impact of SDs on μECoG (i.e., functional status of the penumbra) while still being minimally invasive, and, since placed in the epidural space, do not disturb normal CSF physiology and circulation. This approach allows us to examine stroke evolution and eventually, the therapeutic effects of drug treatment or alternative interventions on stroke outcome.

## Methods
This study was approved by the Duke University Animal Care and Use Committee. A total of 25 rats were used in this study, data from 18 of which were included into the final analysis. Male Wistar rats weighing 280–340 g (age 10–12 weeks) were housed in adequately spaced cages with a 12-hour light/dark cycle with light from 8 am to 8 pm. We have complied with all relevant ethical regulations for animal use.

### μECoG arrays
Figure 1a shows the high-density μECoG arrays over the brain surface with a demonstrated infarct (bland or whitish area). The array contacts were fabricated by depositing gold on biocompatible polyimide or liquid crystal polymer substrates[16–19]. The final thickness of the combined substrate layers was ~50 μm for LCP and 25 μm for the polyimide, thus making the array flexible. Thin wires within the array (12.5–25 μm) enabled high-density interconnections between the metal contacts and the recording apparatus. The gold electrodes were 200 μm in diameter, providing a low-impedance interface (~50 kΩ @ 1 kHz). The arrays have 61 electrodes arranged in an 8 × 8 grid with 420-μm spacing (one electrode is discarded in some of our recordings). The total recorded area is ~3.4 mm × 3.4 mm, sufficient to record both from normal as well as infarcted regions.

Figure 1b shows an awake rat after recovery from the array implantation surgery. The head-mounted interface board was lightweight and allowed the rats to move, groom and feed freely.

A critical requirement was the need to record at a suitably low, high-pass filter setting near DC (i.e., <0.02 Hz) since the cortical spreading depressions have very low-frequency components. Therefore, to augment the lower frequency signal detection we have electroplated the contacts with PtIr (Platinum Group Coatings LLC), reducing the impedance of the interface to ~ 1-2 kΩ @ 1 kHz. Figure 1c shows a graph of frequency vs impedance showing electrochemical impedance spectroscopy (EIS, impedance as a function of frequency) of an electrode array after coating.

### Data acquisition system
We previously developed and validated a multiplexed, high-density data acquisition system for neural signal acquisition[17], which has been adapted for near-DC-coupled recordings. We used DC-coupled amplifiers to simultaneously record ultra-slow signals, including SDs, and higher-frequency neural signals. The amplifier gain was reduced to 3×, providing a large input voltage range (+/− 660 mV) while maintaining a low noise level (5 μV rms). Using this system, we can simultaneously record both ultra-slow high-level signals, including those near DC, and low-level neural signals, such as ECoG, concurrently, in awake, freely moving animals[17]. A schematic diagram is shown in Supplementary Fig. 1. The multiplexed outputs are recorded using custom data acquisition system (DAQ) utilizing commercial multifunction data acquisition cards (PXI-6289, National Instruments Corp).

As in our prior reports[17,18,20], 60-ch electrode outputs were passed through a 61-pin zero-insertion-force (ZIF) connector (Hirose Electric Co., LTD.) to a head-mounted interface board. During each recording session, an analog, multiplexing headstage was connected to the head-mounted interface board to buffer, multiplex, filter, and amplify the 60 neural signals (one electrode was not recorded). The analog multiplexed outputs were then connected to a remote data acquisition system using an ultra-flexible μHDMI cable (Draco Electronics, LLC, CA, US).

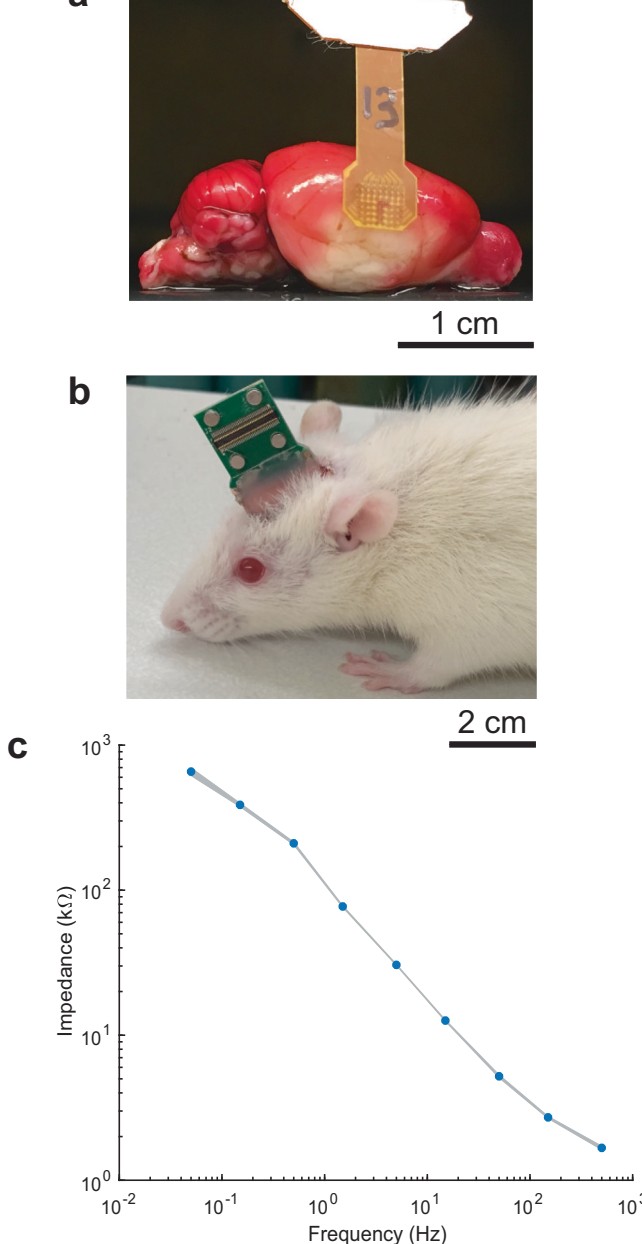

**Fig. 1 | Custom, DC-coupled recording system for long-term recording of SDs in awake, freely behaving rats. a** Typical MCA territory infarct (white area) revealed by TTC histology at 24 hours after 60 minutes ischemia, with the array super-imposed over the cortex, near the corner of the stroke core. The recording area of the array is 3.4 × 3.4 mm and can cover regions of both the penumbra and adjacent healthy brain tissue, depending on placement and stroke involvement. **b** Awake rat with an implanted μECoG array and adapter board. **c** Electrochemical impedance spectroscopy (EIS) of a PtIr electrode. Blue data points represent the mean; width of the grey shadow represents the standard deviation. Deposition of PtIr on the gold electrode drastically lowers impedance.

The analog multiplexed signals were carried on the shielded, twisted pairs of a μHDMI cable, and measured differentially by the remote DAQ. These filtered and amplified signals were sampled at 125,000 Hz with an oversampling factor of 16 on two PXI-6289 data acquisition cards, yielding a multiplexer switching rate of 7812 Hz. The multiplexers combined 15 electrode channels to one output. Samples taken before the multiplexed output had completely settled to 18-bit accuracy were discarded. The remaining samples were averaged to reduce noise, yielding an effective final sampling rate of ~521 samples per second per electrode channel. The combination of the head-mounted interface board and the multiplexed headstage had a height profile of ~36 mm and weighed ~2 g, making it suitable for long-term, awake recording in a freely moving rat. Detailed system wiring diagram along with the functional schematic block diagrams are shown in Supplementary Fig. 1.

### Neural signal analysis
To remove unwanted offsets from the recorded signals, a high-pass, first-order Chebyshev Type I digital filter with cutoff 0.005 Hz was used. For further analysis of specific local field potential (LFP) frequency bands, a lowpass, first-order Chebyshev Type I digital filter was used (with cutoffs of 0.5, 20, or 200 Hz). For frequency analysis, a multitaper power spectral density estimate was used. All analysis was completed in MATLAB or Python.

### Implantation of μECoG array for long-term awake recordings
Rats were anesthetized with isoflurane (induction 5%, maintenance 1.5%), and the head was secured in a stereotaxic frame. After an incision to expose the skull a 5-mm x 2-mm slit craniotomy was made over the right hemisphere to expose the brain, and the sterilized electrode array was slid into the epidural space over the parietal-temporal cortex (Fig. 1a), and then encapsulated. A silver wire was soldered between the headstage and 5 metal (uncoated) skull screws for grounding and as a reference. The μECoG array was connected to the recording system via a tethered ultra-flexible μHDMI cable in the home cage of the animal.

### Acute in vivo setup
For initial evaluation and testing of the hybrid array, a full cranial window preparation (a 5 × 5-mm) was performed. Concurrently with the placement of the array, SDs were induced with KCl and the propagating signals associated with SDs measured with glass intra-parenchymal recordings through a DC-coupled amplifier (filled with 0.2 mM NaCl, ~2 MΩ, referenced to a distant AG-AgCl balanced ground). The depth of the intraparenchymal recording within the cortex was 200 μm. These initial experiments allowed a direct comparison of a traditional SD event, as recorded with true DC parenchymal glass electrodes, to the same event simultaneously recorded with the surface array.

### Array characterization in vivo for acute experiments under anesthesia
To demonstrate the feasibility of detecting SDs using the μECoG array, we performed a 5 × 5-mm craniotomy in both healthy rats (n = 2) *and after permanent MCAO* (n = 3) under anesthesia with full physiological monitoring. We then implanted the μECoG array into the epidural space and either initiated SDs by applying topical KCl, or inducing cortical damage with an MCAO stroke (representative example shown in Supplementary Fig. 3), then monitoring spontaneous SD events. All SDs occur as a continuum, regardless of the mode of induction[12,21]. The features of SDs (i.e., morphology) are dependent on multiple factors, including the state of the tissue (healthy or not) they are propagating through, the site of recording with respect to the cortex (i.e., parenchymal vs superficial to the cortex), the nature of the electrodes and associated grounds (i.e., metal vs ionic, balanced vs. unbalanced), rather than simply on the mode of induction. Therefore, any distinction between KCl and MCAO-induced SDs was not necessary when evaluating the performance of the array during acute experiments. We simultaneously recorded signals from a glass microelectrode placed into the brain parenchyma (200 μm depth) at the border of the μECoG array to confirm the characteristic waveforms associated with each SD and tracked the parallel progressive movement of SD events across the brain. Rats did well after the chronic implantation. The arrays and recording electronics are lightweight and did not impair grooming and feeding.

### Induction of focal cerebral ischemia

In chronic experiments, three days after the implantation of the µECoG arrays, rats underwent middle cerebral artery occlusion (MCAO, n = 20, 7 technical pilot experiments excluded from data analysis)[22]. Male Wistar rats weighing 280–300 g (10–12 weeks old) were anesthetized with 5% isoflurane in 30% $O_2$ and 70% $N_2$, endotracheally intubated, and mechanically ventilated with 1.5% isoflurane in 30% $O_2$/70% $N_2$. Rectal temperature was maintained at 37.5 °C ± 0.2 °C by surface heating or cooling during anesthesia. Rats were prepared for middle cerebral artery occlusion (MCAO) as previously described[22,23] with modification. A midline ventral cervical skin incision was made, and the right common carotid artery was identified. The external carotid artery was then isolated, ligated, and divided, and the internal carotid artery was dissected distally until the origin of the pterygopalatine artery was visualized. To achieve MCAO we used commercially available nylon monofilaments (Doccol Corporation, Sharon, MA, USA) with 0.31-mm diameter silicon tips, inserted the coated tip into the external carotid artery stump, and advanced it 19–20 mm from the carotid artery bifurcation into the internal carotid artery. The filament was removed after 60–90 minutes MCAO (transient MCAO) to allow reperfusion. Wounds were closed, isoflurane was discontinued, and rats were extubated. Immediately after anesthesia emergence, a screening neurologic assessment was performed using a 4-point scoring system (Bederson score)[24]. This assessment includes forelimb flexion, resistance to lateral push, and circling behavior, and the scoring scale (0–3) reflects basic neurologic deficits, confirming successful stroke.

### Statistics and reproducibility

Two types of statistical analyses were conducted: ANOVAs and cluster-based nonparametric permutations. ANOVAs were conducted when comparing power spectra across different conditions, while cluster-based analyses were conducted when comparing voltage heatmaps at two different time points.

ANOVAs were conducted in JMP Pro Version 15 (SAS Institute Inc., Cary, NC). To protect against multiple comparisons, we first ran global ANOVAs and conducted follow-up tests only if the relevant interaction terms were statistically significant (alpha = 0.05). We used the Tukey Honestly Significant Difference (HSD) as a post-hoc test when appropriate (i.e. when there was a statistically significant effect, alpha = 0.05). Therefore, all reported p-values are protected from multiple comparisons. A detailed overview and the results of our statistical analysis can be found in the Supplementary Information.

Cluster-based nonparametric permutation tests were conducted in MATLAB R2019b (MathWorks, Natick, MA). We used the MATLAB function "permutest" that is peer-reviewed and available on MATLAB Central File Exchange[25,26]. We used cluster-based permutation tests to identify statistically significant (alpha = 0.05) clusters in our voltage heatmaps comparing post-stroke to baseline conditions. "Permutest" corrects for multiple comparisons arising from the high number of LFP recording channels in our arrays. A detailed overview of how we applied cluster-based permutation tests can be found in the Supplementary Information.

All animals recovered from the array implantation surgery without complications (such as infection around the head-mounted interface board) within 48 hours. All animals exhibited the anticipated physical deficits after transient MCAO, although the severity of the stroke varied across animals. Variable stroke severity was a desired feature of our experimental design, as we wanted to track cortical dynamics across a heterogenous population of ischemic animals.

### Reporting summary

Further information on research design is available in the Nature Portfolio Reporting Summary linked to this article.

## Results

Clarification of terms used in our report, which relate to a comparison of our chronic signals recorded from superficial to the cortex (in both our animal situation and in human clinical situations:

1. Spreading depolarizations (SDs) waveforms can be detected in healthy brain tissue, typically elicited by KCl. These events are typically extracellularly negative responses up to 20 mV (as measured with ionic, glass intracortical penetrating electrodes referenced to a balanced ionic distal ground), with a duration of 60–180 s. These intracortical waveforms are accompanied by a concurrent depression of the ECoG in the involved regions.

2. Spreading depolarizations can also arise spontaneously, triggered in the penumbra of an infarct, but then spreading into normal brain. These ischemic SD events exhibit various shapes and durations depending on their capability for spread and underlying tissue status. Once an SD is initiated in the ischemic penumbra, it may spread into all tissue that can respond, which includes adjacent healthy tissue. In healthy tissue it will display classic SD characteristics once initiated (depending on the recording location). However, severely ischemic or damaged tissue (i.e., infarct core) cannot sustain SD propagation, so SD events do not invade such tissue. ECoG depression is present in tissue that is still functionally viable, while background ECoG is depressed at baseline in tissue that is already severely compromised by an infarct.

3. There are multiple waveform differences between the typical, ionic, glass intracortical electrode recordings (coupled to a DC amplifier) and metal electrodes above the cortical surface coupled to near-DC recording amplifiers, when referenced to unbalanced, metal skull screws. These differences likely result in different waveform morphology of recorded SD events compared to the intra-parenchymal recording setup.

4. Low-level spontaneous ECoG activity represents the dynamic state of the underlying brain. For example, in healthy regions in awake brain spontaneous alpha and beta rhythms occur, whereas over stroke regions the ECoG is either silent or has excessive low frequency (i.e., delta) activity.

To facilitate detailed monitoring of SD characteristics across the brain, the concurrent measurement of brain activity (ECoG) for monitoring of both SDs (at the mV level), and the highly dynamic, inhomogeneous stroke boundaries defined by spontaneous ECoG (at the µV level), we modified previously developed hardware to achieve that goal[16–19]. The application of this modified recording system to the field of stroke is innovative as it includes: 1) a wide sampling of cortex across stroke boundaries and normal tissue; 2) multiplexed, DC-coupled amplifiers with low gain to detect the SD events; 3) a high resolution analog to digital converter (ADC) with sufficiently low noise to detect ongoing ECoG signals which vary between stroke regions and normal brain areas; 4) long-term recordings with the animal awake and tethered using a thin (2 mm) and flexible cable, to facilitate as normal of behavior as possible.

### Electrode array

We used flexible, biocompatible, high-density µECoG arrays that have been previously fabricated and extensively validated[16–18]. Platinum iridium (PtIr) was electroplated on the gold contacts to further reduce the impedance of the interface by about an order of magnitude (Platinum Group Coatings LLC). PtIr electrode contacts demonstrate impedances ~600–700 kΩ at 0.05 Hz.

### Hardware

Neural potentials from each electrode contact were buffered by a unity-gain amplifier with ~1GΩ input impedance (TI OPA2376) prior to 16:1 multiplexing. The reference electrode assembly consisted of five bone screws placed across the skull thickness (into the epidural space) and then ganged together, sufficiently deep to touch the dura, and connected via silver wire

among each other and then to the headstage ground. The bone screws also recorded neural signals from the underlying brain regions, though any common signal among all five bone screws would be averaged. We modified the headstage amplifier for DC-coupled recording by removing the high-pass filter and reducing the gain to 3×, achieving a wide voltage input rage of ± 660 mV to accommodate varying DC electrode potentials. At the same time, by utilizing a high-resolution (18-bit) ADC (NI PXI-6289), this system maintained a low noise level of 5.08 ± 0.21 µV rms in the relevant neural signal frequency ranges of 2 – 200 Hz (used to evaluate suppression of ongoing brain activity during SD events). We calculated the DC drift rate of our recording system in a fully implanted and freely behaving rat (Supplementary Fig. 2). The DC drift rate was between -0.89 and 0.86 mV/min. The DC drift rate is well below the average amplitude of SD events we observed in our experiments (both acute and chronic preparations) and therefore does not interfere with SD recordings. These amplifier changes allowed us to record near-DC signals from the brain, particularly spreading depolarizations (SDs), and concurrently map dynamic stroke evolution with higher frequencies. Further details of the recording hardware are described in the Methods section.

## Noise levels, ambient ECoG and SD recordings

Simultaneous monitoring of large amplitude, low-frequency extracellular spreading depolarizations concurrent with low amplitude, higher frequency spontaneous brain activity (ECoG) in freely moving individuals poses unique methodological challenges.

Cortical activity in the EEG frequency range reflects functional activity occurring at frequencies up to 200 Hz and is usually recorded with AC-coupled amplifiers, which filter out slow and DC (i.e., <0.1 Hz) potentials. SDs, however, induce a slow negative waveform (<0.1 Hz). The limitations of traditional AC-coupled techniques for SD monitoring have been partially overcome by the use of amplifiers with higher time constants, i.e., lower high-pass filter cutoff frequencies[27]. In contrast, it is possible to make unfiltered, full-band recordings with DC-coupled amplifiers from platinum electrodes in patients, though these have considerable drift, limitations of metal electrodes, and lack of DC stability[27]. DC penetrating recordings are the gold standard in animal studies, typically performed with glass electrodes filled with ionic solutions containing Cl- (and an internal pickup Ag/AgCl wire) and referenced to a Ag/AgCl distant ground in the neck[7]. The

ionic media provides full mobility of charge, compared to metal electrodes. However, both, recent clinical studies as well as classic DC recordings in animals are restricted in important ways: 1) the recording is limited to very few recording sites (i.e., 1–2 intra-cortical glass micropipettes in animals or 4–6 PtIr electrode contacts in humans; and 2) animal experiments require anesthesia for the recordings, substantially limiting experimental time to a few hours.

We have addressed and solved these problems:

1) we can reliably insert the arrays into the epidural space over the cortex through a small slit craniotomy, as well as record from the epidural space, since our arrays are sufficiently thin to be placed between the skull and the dura; 2) there are 60 electrode contacts, providing a high density of recording sites over the cortical surface; and 3) recordings are performed through a previously validated high-density data acquisition system for neural signals[28,29], which has been adapted for near DC-coupled recordings of ultra-slow signals from the brain, including SDs. Using this system, we can simultaneously record ultra-slow signals and neural signals concurrently in awake, freely moving animals over long periods of time.

Figure 2 shows representative recordings from awake, implanted animals at specific frequency bands, to illustrate that our recording setup has low noise levels (and thus high signal-to-noise ratio, SNR, Fig. 2b) for the frequency range 0–200 Hz. We also reliably recorded low-frequency cortical fluctuations (Fig. 2a, 0.005–0.5 Hz) together with spontaneous EEG-level brain activity (Fig. 2a, 0.5–20 Hz). An ANOVA on LFP power identified a statistically significant interaction ($p < 4e-9$) between recording type (pre-MCAO, post-MCAO, and post-mortem after euthanasia) and frequency band (0.005–0.5 Hz, 0.5–20 Hz, and 20–200 Hz). Therefore, we subdivided the data by frequency band and conducted Tukey HSD post hoc tests. Across all frequencies, post-mortem (i.e., after euthanasia) LFP power is lower than LFP power during pre-MCAO and post-MCAO. Low frequency (0.005–0.5 Hz) power is highest in the post-MCAO condition, which is expected because SDs are present. Middle-frequency power (0.5–20 Hz) is highest in the pre-MCAO condition, which is expected because during SDs (which are present post-MCAO) there is suppression of ongoing activity. All reported post-hoc tests were statistically significant ($p < 0.03$). A complete statistical report (F ratios, confidence intervals, and p values) can be found in the Supplementary Notes 1.

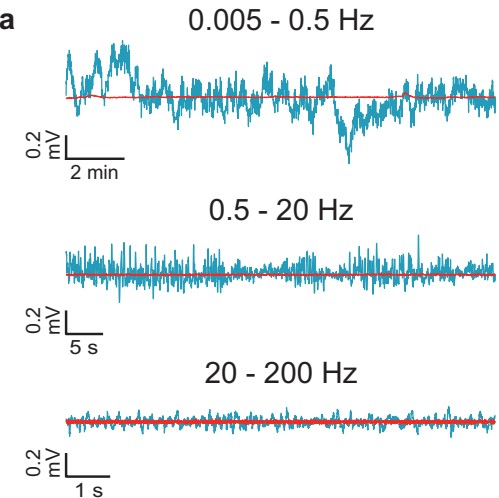

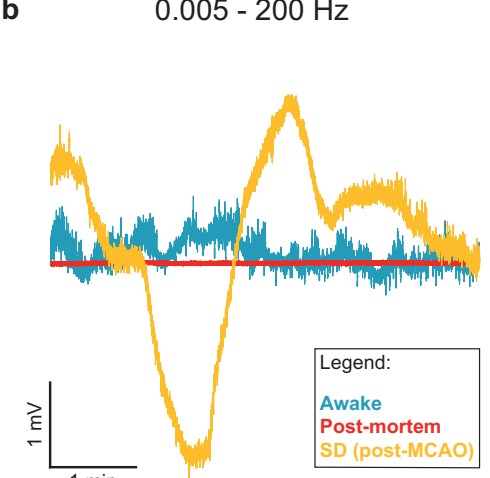

**Fig. 2 | The DC-coupled recording system exhibits low noise and can reliably detect both low frequency signals, such as SDs, as well as higher frequency, ongoing µECoG activity. a** Time series data from a representative channel of an awake, freely behaving rat and a post-mortem recording, divided into three frequency bands (0.005–0.5 Hz, 0.5–20 Hz, and 20–200 Hz). The postmortem recording is indicative of system noise alone with no neural signals. Neural signals in all three frequency bands are larger amplitude than, and thus can be distinguished from, system noise. **b** Broadband time series data from a representative channel of a rat pre-MCAO, post-MCAO, and post-mortem. SD amplitude is well above system noise, and its time course characteristically different from that of random noise. ANOVA and post hoc Tukey HSD determined that LFP power was statistically different ($p < 0.03$) across all three frequency bands (0.005–0.5 Hz, 0.5–20 Hz, and 20–200 Hz) and all three recording types (pre-MCAO, post-MCAO, and post-mortem).

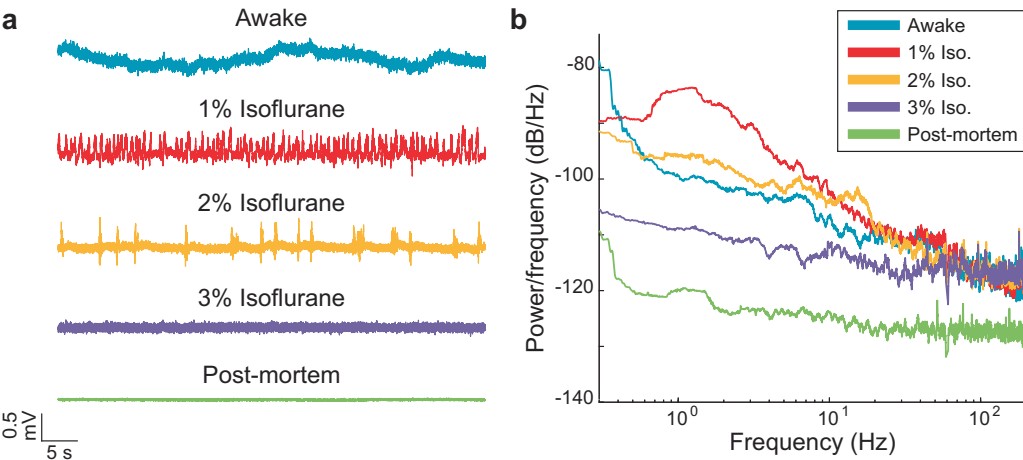

**Fig. 3 | The μECoG array reliably distinguishes different brain states of a freely behaving rat. a** Voltage time series from a representative channel of a rat while awake, under different levels of anesthesia (1%, 2%, and 3% isoflurane), and post-mortem. **b** Power spectra of the time series in **a**. ANOVA and post hoc Tukey HSD determined that delta activity (1–4 Hz) was statistically different ($p < 5e$-8) across all pairs of brain states (except one). Delta activity was not statistically different between awake and 2% isoflurane.

## Two parameter analysis: ECoG and SD detection

Next, we performed specific experiments to determine if each of the important, critical parameters (i.e. continuous ECoG assessment and SD events) were reliably detected and processed by our recording system and subsequent analysis.

## ECoG recordings in different (well-known) brain activity states

We varied anesthesia levels to evaluate the sensitivity of our system to detect ECoG changes, since anesthesia reliably alters ECoG signal characteristics[28–30]. Figure 3a demonstrates voltage traces over time at different anesthesia depths, including the recording system baseline at the bottom after euthanasia. These example recordings show intermittent delta activity (at 1%), burst suppression (at 2%) and then a quiet ECoG (at 3%), which are higher in amplitude than the low voltage of the dead condition (lowest trace). Power spectra differentiated these various brain states, including awake, 1%, 2%, 3% isoflurane, and dead (at the bottom), as expected (Fig. 3b).

An ANOVA identified a statistically significant effect ($p < 4e$-13) of brain state on the LFP power in the delta band (1–4 Hz). At 1% isoflurane, there is an increase in delta activity compared to the awake, baseline recording; 2% isoflurane suppresses this activity back to baseline; 3% isoflurane further decreases this activity below baseline. Postmortem is statistically distinguishable from all alive recordings. Delta activity was statistically different across all pairs of brain states except awake and 2% isoflurane. All reported post-hoc tests were statistically significant ($p < 5e$-8). A complete statistical report (F ratios, confidence intervals, and p values) can be found in the Supplementary Notes 2.

Since we anticipated recording ECoG frequency differences between the stroke core/penumbra and healthy regions, this experiment confirmed the ability of our system to detect those changes[31,32]. Our array and recording system were therefore able to monitor both SD events (i.e., Fig. 2b) and ongoing, spontaneous brain ECoG activity (Fig. 3), in a variety of conditions.

## SD detection: comparison against the gold standard

We next wanted to directly compare intra-cortical DC recordings with the array responses for detection of SDs. In order to validate that our system can reliably detect SD events we performed in vivo experiments in anesthetized rats, recording from an array placed directly over the cortex (in the epidural space) together with a confirming, intracortical glass DC electrode (Fig. 4a shows the array and electrode location). We demonstrate the detection of a spontaneous SD event (after MCAO stroke) simultaneously in both the penetrating electrode in the cortex (Fig. 4b) and the μECoG array recordings

from the surface (Fig. 4c). Our arrays were clearly able to detect the same SD event as the glass recording electrode. However, the array also recorded the spread and conduction of the SD event across the cortex, as shown in Fig. 4c, with some delay between the contacts on the array as expected. Additionally, a top-down reconstruction of the array voltage responses (Fig. 4d and Supplementary Video 1) shows the spread of the SD event from the anterior medial aspect of the cortex to posteriorly on the cortex, validating that the array has sufficient resolution to detect the slow (~6 mm/min) SD conduction across the cortex. We confirmed the consistency of simultaneously recording a SD response from the array and the penetrating electrode in 5 independent acute rat experiments (two KCl experiments with induced SDs and three middle cerebral artery stroke experiments with associated, spontaneous SD events). We picked up all KCl-induced SDs but missed three small SDs in the stroke experiments due to limited spread and low amplitude, <1 mV. Within a total of 850 minutes recording time across these 5 animals we detected 47 SDs in the intra-cortical electrode and 44 in the array, indicating a 93% detection rate.

## Complexity of SD shapes and reliability of detection

With the progression of ischemia, post-stroke SD waveform shapes and conduction become more heterogeneous in amplitude and duration over time, compared to the homogenous SD events induced by KCl. Hence, as described above, we analyzed longer-term recordings over 40 min for detection of SDs after a MCAO stroke, in anesthetized animals ($n = 3$, total recording duration 560 minutes, 25 SDs in the gold standard, 22 SDs in the array). Figure 5 shows a typical acute post-stroke recording sequence over 40 minutes, illustrating the diversity of SD waveform shapes and durations. Figure 5a shows the location of three representative array channels and the intraparenchymal electrode with respect to the skull. Our system is able to detect most SDs of various shapes and amplitudes similar to that of the penetrating electrode, but occasionally with complex SD waveform (i.e., first 3 SD events between 3 and 12 min). The penetrating glass electrodes use a consistent but indifferent, balanced, neck Ag-AgCl ground, away from the cortex, with a consistent negative response. In contrast, the metal array contacts are referenced to a mixed ground assembly of metal skull screws spread across the cortex, so there may be cortical activity reflected within the ground that influences the polarity of the event recorded from the array as well as the influence of the unbalanced metal contacts and references. Moreover, epidural recordings of spreading depolarizations in patients (also with metal electrodes and grounds) can in fact show a positive waveform phase[33], agreeing with our data. Since the array covers a large area of brain compared to the single penetrating electrode, every channel records a

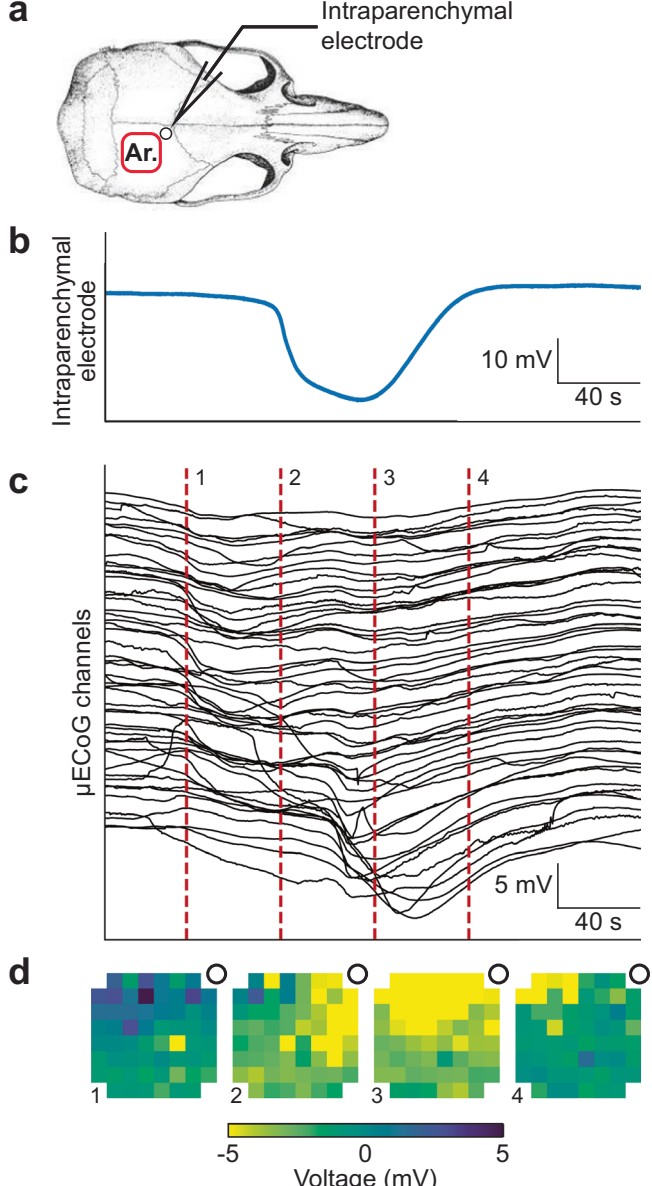

**Fig. 4 | Concurrent recordings from an intraparenchymal electrode and the high-density array of a rat post-MCAO under anesthesia. a** Location of the array and the intraparenchymal electrode with respect to the skull. The depth of the intraparenchymal electrode was 200 μm. **b** Intraparenchymal electrode recording of a SD. **c** Array recording of the same SD. The voltage time series of each of the 60 channels is shown as a stacked series. **d** Voltage heatmaps of the SD, taken at the time points shown in **c**, reveal the spatial propagation of the SD (full animation in Supplementary Information).

different aspect of the SD as well as displays the different tissue status while the SD is propagating through it (as demonstrated by the three representative array channels in Fig. 5). Faulty channels with atypical drift or noise were screened out during baseline recordings (before stroke induction).

The variable, waveform appearance of the SD responses across time and regions of the cortex involved, as recorded from the array, requires a consistent approach to characterizing these responses to optimally detect events. This is also a difficult issue in human studies using subdural metal electrodes[12,27,34] requiring clear criteria for detection of an SD event. However, we have more contacts and cortical coverage than the typical human strip recordings, so the spatial characteristics may also be useful in clear

detection of SD occurrences, despite the ambiguity noted in single recording channels.

It is important to highlight the impact of filtering our raw data in the near-DC range. As illustrated in Fig. 5b, filtering at 0.005 Hz changed the time course of particularly long events (such as the ones at 10 minutes and 30 minutes). Although filtering the data made identification of SDs easier when sorting through large data sets, it also can influence and alter the duration and waveform shapes of SDs. However, we can also refer back to the unfiltered recordings to determine event duration. The analysis pipeline we propose is to identify events using filtered data, and then refer back to the original, unfiltered, near-DC-coupled recordings to characterize further SD characteristics such as duration. These recordings and analysis approaches were replicated and confirmed in $n = 13$ additional stroke animal experiments (duration up to 24 hr. after stroke occurrence), analyzing μECoG brain activity over 154 single SDs and 32 clusters, with a total SD/cluster duration of 1003 minutes.

To further elucidate the dynamic physiology of the penumbra, we analyzed how the root-mean-square (RMS) voltage in the frequency range 0.5-200 Hz varied at two different time points (0 min and 20 min). We analyzed neuronal activity above 0.5 Hz because that is the range of spontaneous ECoG frequencies that are suppressed during SDs. The spatial analysis of RMS voltage at 0 min (before the SDs) revealed the border of the expanding penumbra: the bottom right corner of the array exhibited greater suppression of ongoing activity, suggesting that this tissue is at an increased risk of death (Fig. 5c). The same spatial analysis of RMS voltage at 20 min (during an SD) revealed the widespread suppression of ECoG activity across the array (Fig. 5d). Cluster-based permutation analysis identified a statistically significant ($p < 0.03$) cluster of ECoG suppression across the array between these two time points. This cluster shows the spatial location of greatest ECoG suppression during the SD. Further details about how we conducted this analysis using previously published methods[25,26] can be found in the Supplementary Notes 3.

### Identification of different SD spread patterns

In order to identify and recognize the various SD patterns of origination and spread across the cortex after stroke, we can visualize these events in top-down topographic maps, with alignment shown in Fig. 6a. Top-down array views of SD events during awake recordings after stroke are depicted in Fig. 6b, c. We can identify the stroke (indicated with red border), since the SDs are not able to invade the core stroke region (anterior lateral aspect of the cortex). Additionally, the spontaneous low-level ECoG frequencies show more delta activity in the border of the stroke region. Figure 6b shows one example of a SD starting on the edge of the stroke boundary, then propagating posteriorly and laterally. Figure 6c shows a different example of a SD, which begins near the edge of the stroke and then more diffusely propagates anteriorly and laterally, but again does not invade the core stroke region (for further visualization see Supplementary Videos 2 and 3). These array maps assist to confirm the spread of SD events across the cortical surface and enable a map of the functional status of the tissue at a given point in time.

Though Fig. 5 shows at least 5 mV for many of the SD events in anesthetized animals, in the awake animals SD events were commonly less than 5 mV in amplitude (and at times were as small as 1 mV). We consistently observed that SD amplitudes detected with our arrays were smaller in chronic recordings when compared to acute recordings. We believe this is attributable to a decreasing signal-to-noise ratio when the animal is conscious and freely behaving, as opposed to anesthetized. Under anesthesia there is less background neural activity, as well as often limited SD spread across the cortex (i.e., more localized SD events). We believe these two reasons explain why SDs have a lower signal-to-noise ratio in chronic recordings compared to acute recordings. Our key criteria when identifying SDs that were smaller in amplitude was evidence of spread across the array and suppression of the focal spontaneous cortical signals on involved channels.

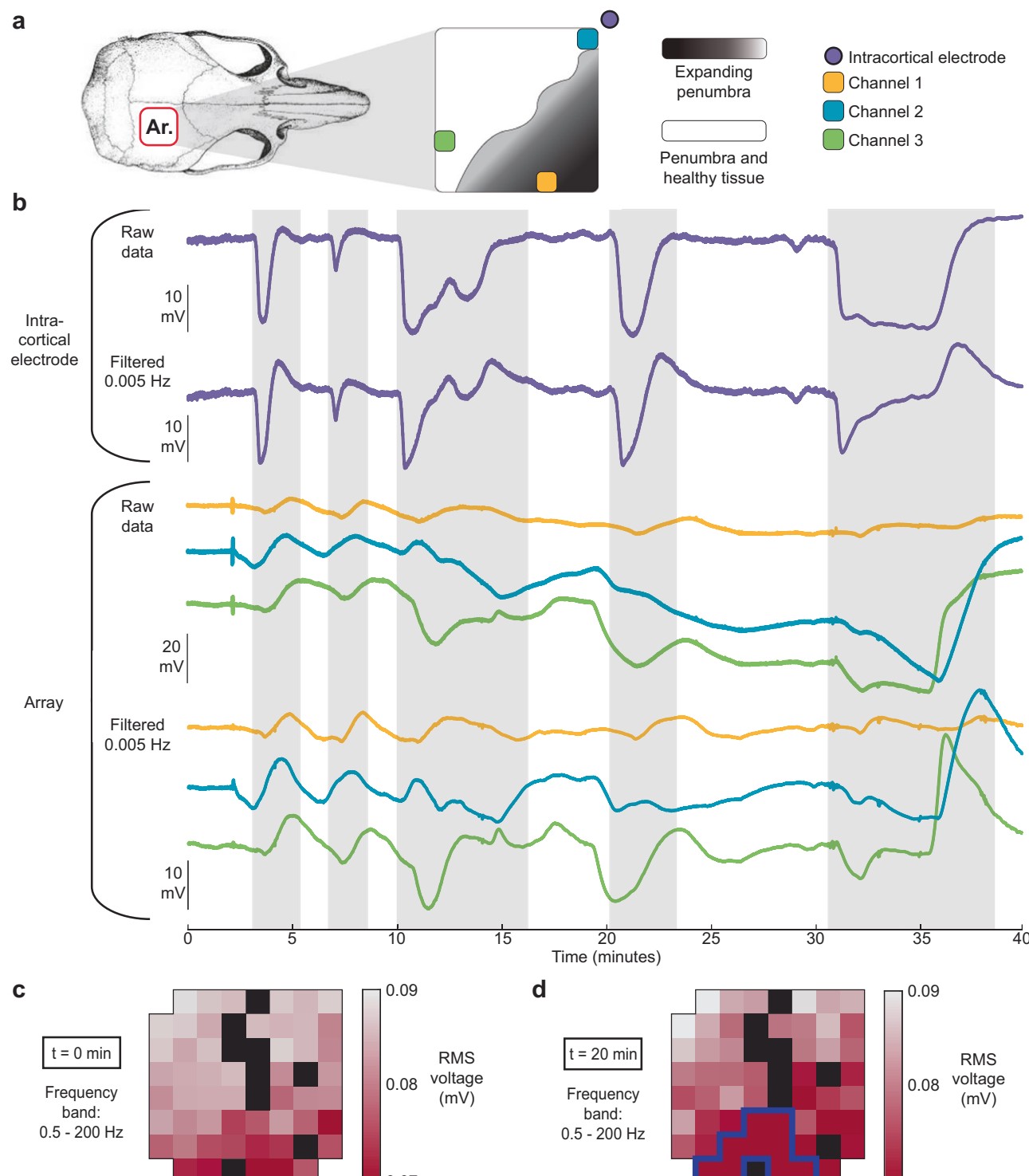

**Fig. 5 | The high-density array successfully detects SDs of varying morphology, as validated by an intraparenchymal electrode under anesthesia. a** Spatial map showing the location of three representative array channels and the intraparenchymal electrode with respect to the skull. The border of the expanding penumbra is also depicted. **b** Raw and filtered data is shown for both the array and intraparenchymal electrode recordings. The array simultaneously detects the same SDs as the intraparenchymal electrode. The true duration of the SDs is preserved in the raw, DC-coupled recordings; however, filtering distorts the morphology of longer-duration SDs.

**c** Heatmap of RMS voltage before the SDs in the frequency range 0.5–200 Hz, revealing the border of the expanding penumbra. **d** Heatmap of RMS voltage during the SDs in the frequency range 0.5–200 Hz, showing suppression of ECoG across the array. In (**c**) and (**d**) blacked-out channels were excluded from data analysis. Faulty channels with atypical drift or noise were screened out during baseline recordings (before stroke induction). The dark blue outline shows a statistically significant ($p < 0.03$) cluster of ECoG suppression across the array between (**c**) and (**d**).

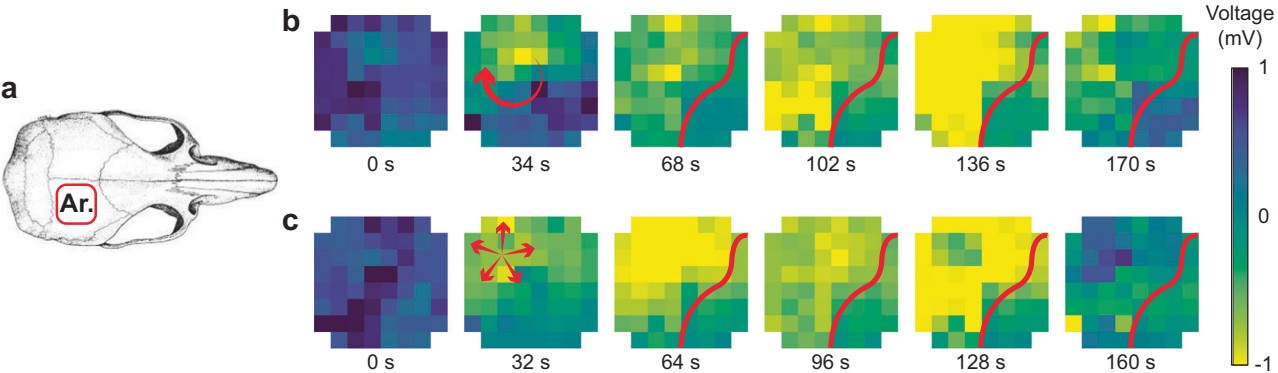

**Fig. 6 | Complex patterns of SD propagation in a long-term recording of an awake, freely behaving rat post-MCAO. a** Orientation of array with respect to the skull. **b, c** Voltage heatmaps during two different SDs reveal complex spreading patterns. Red arrows indicate direction of propagation. Red line indicates the border of the expanding penumbra. SDs cannot propagate into regions of severely injured tissue, particularly the stroke core, hence deviate around these regions (full animation in Supplementary Information).

## Concurrent recording of SDs and ECoG

A critical capability of our recording system is to be able to both simultaneously detect higher-level SD events and concurrent, low-level spontaneous brain activity, in order to map ischemic boundaries. With each SD the ongoing cortical activity (ECoG) is suppressed due to profound cellular depolarization and inactivation[10], hence the focal suppression is a critical aspect of SD detection. In Fig. 7 we display a short sequence of SDs occurring 4 days after MCAO in a representative awake recording. We display data from 6 representative array channels. Figure 7a shows the relative location of these channels with respect to the skull, and Fig. 7b shows the SD waveforms at each channel. Note that there are variable negative waveform shapes depending on the channel location near or away from the likely stroke boundary (as indicated in the upper diagram by a shaded dark area). Near the stroke (or within) there is a minimal SD response, whereas in more normal areas (i.e. red channel) there is a large response. As shown by the ECoG frequency responses (Fig. 7c), the first SD event led to a minimal ECoG suppression, whereas the second two SDs resulted in a more profound ECoG suppression, similar to human recordings. During the third SD there is widespread suppression of ECoG activity across the array (Fig. 7e) compared to before the cluster of SDs (Fig. 7d). Thus, depending on the location of the specific array contact (i.e., core vs. penumbra vs. healthy) there may be more or less SD negative deflection and secondary ECoG suppression.

The SD events shown in Fig. 7 occurred 4 days after MCAO. To determine whether our array could detect the expanding border of the penumbra over time, we compared the RMS voltage heatmaps between baseline (when the animal was awake and before MCAO) and 4 days after MCAO (Fig. 7d). Cluster-based permutation analysis identified a statistically significant ($p < 0.05$) cluster of ECoG suppression across the array between these two time points. This cluster shows the spatial location of greatest ECoG suppression during an SD. Further details about how we conducted this analysis using previously published methods[25,26] can be found in the Supplementary Notes 3.

Over time further ECoG suppression may not be detectable as brain activity is progressively reduced due to the deterioration of tissue status. As discussed in Harting's review[12], the cortical regions showing spreading depolarizations associated with ECoG suppression may display progressive deterioration of tissue status, with loss of ECoG activity associated with ongoing SDs, and finally continue into a terminal depolarization with no recovery. Furthermore, ECoG activity in itself is an important indicator of tissue status after injury. Direct electrocorticographic (ECoG) patterns are characteristic during stroke phases, showing both increased power in low (delta) frequencies and the decreased power in high (alpha) frequencies that arise as a result of an acute stroke[31,35].

## Synthesized criteria for SD identification and stroke boundary assessment

Based on these correlative acute recordings and longer-term summary results we can formulate criteria to define presence and occurrence of SDs, particularly if they involve spread across several contacts, for objective analysis of the array recordings. An exemplary recording is depicted in Fig. 8, accompanied with power spectra at three different time points, revealing the spatial progression of ischemia. An ANOVA and post-hoc Tukey HSD tests were able to identify channels located in the penumbra (where the tissue is still viable at the beginning of the recording) versus channels that represent more severely injured tissue. Channels 1 and 2 (which exhibit large amplitude SDs) are in the penumbra and show viable ECoG activity at the beginning of the recording. Over time, this activity decreases. On the other hand, Channel 3 represents more severely injured tissue with greatly reduced ECoG activity at the beginning of the recording. Over time, this activity does not recover. All reported post-hoc tests were statistically significant ($p < 0.002$). A complete statistical report (F ratios, confidence intervals, and p values) can be found in the Supplementary Notes 4.

We have developed the following criteria to define SD characteristics and to delineate apparent stroke regions based on data from 13 individual animals, evaluating μECoG, 154 single SDs and 32 clusters with a total SD duration of 1003 minutes:

1. SD occurrences are distinct events larger than 1 mV (of either polarity), in a minimum of three channels, lasting for at least 60 seconds.
2. SD spread: a) SDs do not invade the core region of the infarct, (because the cells inside the core are already permanently depolarized) effectively demarcating channels representing the core; b) SDs show varying shapes and spread patterns in the surrounding regions; and c) classical SD characteristics can identify spread into residual, healthy tissue. Topographic maps can visualize these spread patterns, demarcating the penumbra and likely regions of origin.
3. Many of the channels involved with the larger SD events also simultaneously show ECoG suppression. Spectrograms highlight that ECoG activity is suppressed during SDs but recover quickly after the SD terminates. The time course of suppression and recovery of ECoG activity may distinguish areas of impaired viability versus areas that display normal ECoG.
4. ECoG spectrograms allow discrimination of representative ECoG frequencies and patterns typical for the core area and penumbra of the infarct, helping to ascertain dynamic stroke boundaries over time.

## Discussion

We demonstrate feasibility of implanting and recording from high-density cortical arrays with 60 electrodes in awake animals, for monitoring both

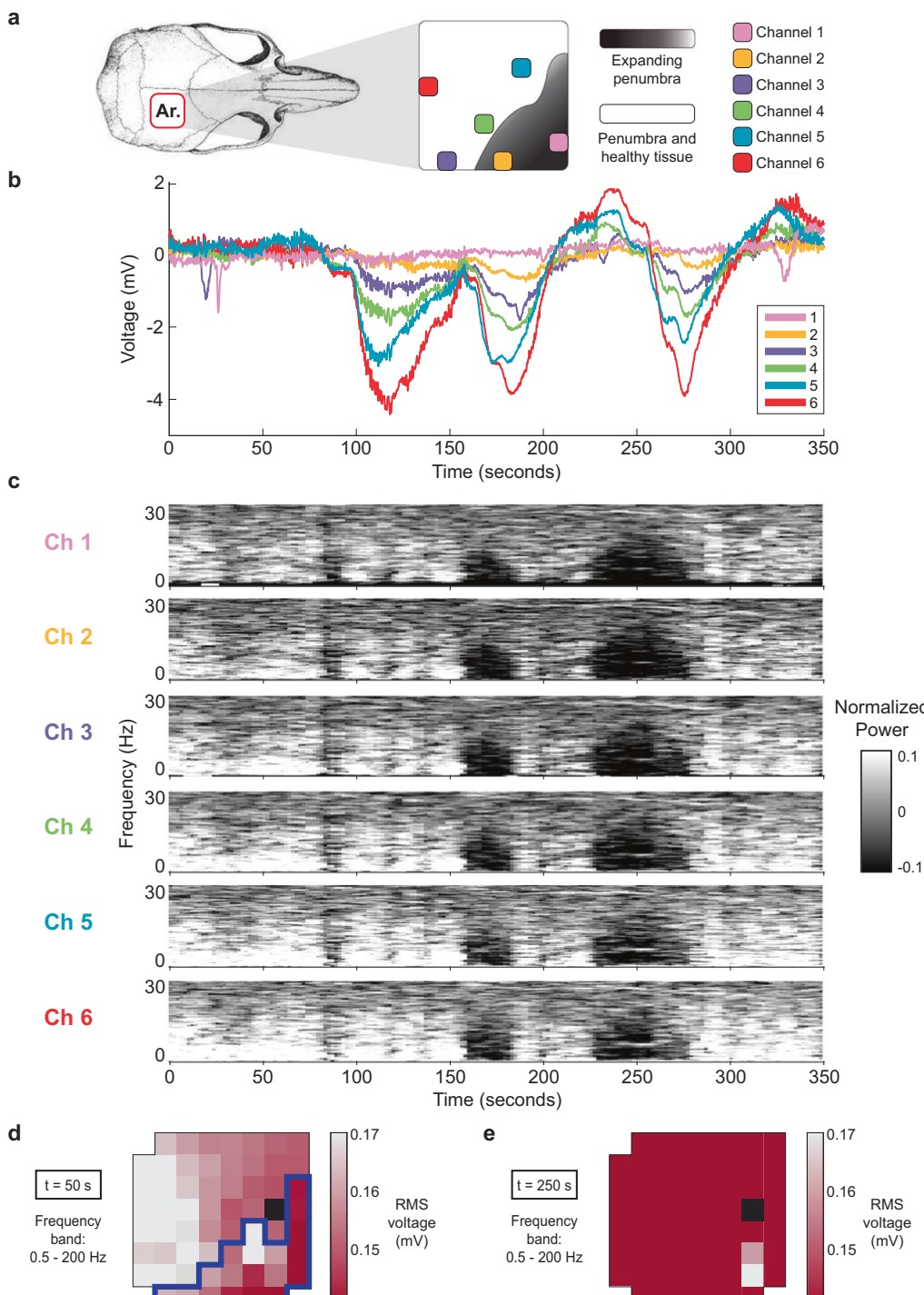

**Fig. 7 | SDs are concurrent with μECoG suppression and the high-density array reveals the border of the expanding penumbra in-vivo. a** Spatial map showing the location of six representative channels with respect to the skull, as well as the border of the expanding penumbra. **b** Voltage time series of the six channels, exhibiting a cluster of three SDs of varying amplitudes. This recording occurred on the fourth day post-MCAO. **c** Spectrograms of the six channels, normalized to a baseline (pre-MCAO) recording. Spectrograms reveal suppression of μECoG during SDs (greater suppression for each subsequent SD in the cluster) and recovery after repolarization. **d** Heatmap of RMS voltage before the SDs in the frequency range 0.5–200 Hz, revealing the border of the expanding penumbra. The dark blue outline shows a statistically significant ($p < 0.05$) cluster of ECoG suppression across the array between baseline and (**d**), i.e. before and after stroke. **e** Heatmap of RMS voltage during the cluster of SDs in the frequency range 0.5–200 Hz, showing suppression of ECoG across the array.

low-level ECoG activity and higher-level SD occurrences after MCAO stroke. We present data acquired from acute experiments favorably comparing cortical DC glass penetrating electrodes with the array channels, to confirm near-equivalent recordings of both induced SD events and patterns of propagation. Further, we demonstrate that SDs can be detected over time

after MCAO stroke in awake animals. Based on these results we have developed criteria (much like in the limited human setting[36]) for presence of SDs, to be able to automatically analyze long periods of data to identify stroke regions and dynamic evolution of stroke boundaries in relation to SD occurrences. These advances will be very useful in characterizing SD

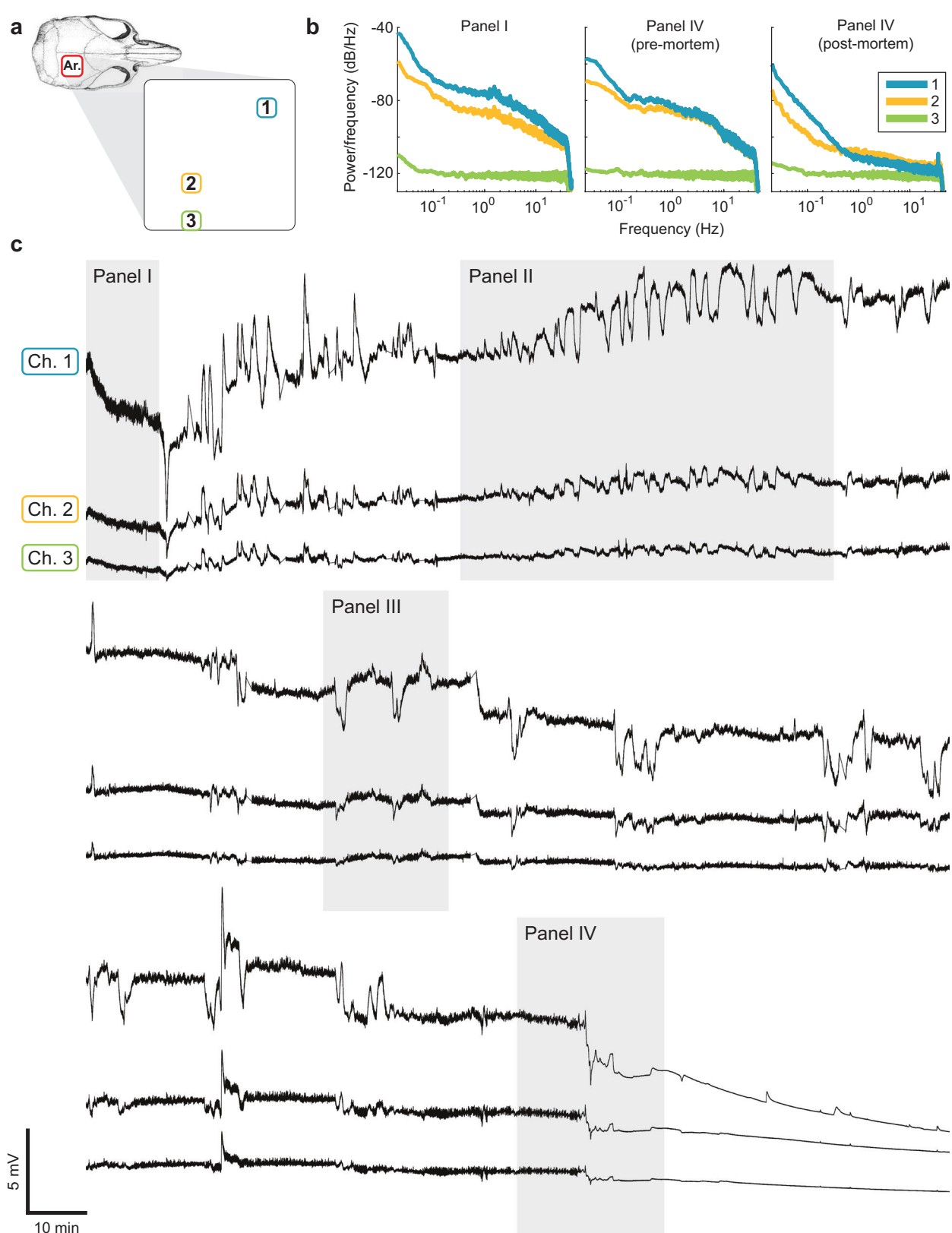

occurrence, dynamic stroke evolution, and potentially treatments in this animal model, and could eventually be translated to humans as advanced monitoring techniques become available.

Spreading depolarizations in healthy tissue are observed as large ultraslow negative potential change in the near-DC frequency range (less than 0.05 Hz) of the ECoG. This SD waveform originates from severe depolarization intracellularly in neuronal soma and dendrites as well as astrocyte involvement[37]. Because the SD waveform is generated by neuronal and astrocyte depolarization combined, it is observed not only in vivo but also in brain slices. Another hallmark of SD occurrence

**Fig. 8 | DC-coupled, long-term recordings of an awake, freely behaving rat showing typical progression of ischemia.** The recordings display three representative channels within a complete recording over 8 hours after a severe stroke that resulted in death of the animal. **a** Spatial map showing the location of three representative electrodes with respect to the skull. **b** Power spectra of the three channels at the beginning and end of the recording, as well as after death, revealing the spatial progression of ischemia. ANOVA and post hoc Tukey HSD determined that the LFP power across all three frequency bands (<0.5 Hz, 0.5–20 Hz, >20 Hz) steadily declined over time for Channels 1 and 2 ($p < 0.002$), but not for Channel 3. **c** Concurrent DC-coupled recordings from the three representative electrodes. Each panel was mean-subtracted when joining consecutive panels. Channels 1 and 2

exhibit large amplitude SDs, representing the penumbra, with still viable μECoG at the beginning of the recording, deteriorating over time. Channel 3 represents more severely injured tissue that initially exhibits small amplitude SDs and depressed μECoG. Over time, as ischemia progresses, the area gets incorporated into the core of the infarct, as SDs fail to invade this core region and μECoG signals are progressively lost. The highlighted panels show typical μECoG and SD behavior: baseline μECoG in channels 1-3 indicating different μECoG levels (I), long cluster of SDs (II), individual SDs exhibiting suppression of μECoG during the SD, and recovery of μECoG after the SD (III), with terminal depolarization that does not recover and sustained, full suppression of μECoG, corresponding with death (IV).

is its slow spread across the cortical surface and hippocampus, at ~3–6 mm/min.

During brain ischemia, when ATP is persistently depleted from the tissue, ATP-dependent membrane pumps such as the Na, K-ATPase fail to restore the membrane potential. Severe metabolic stress imposed by stroke-induced spreading depolarizations arises if there is insufficient collateral for neurovascular coupling and increased blood flow. Hence, spreading depolarizations (SDs) are a major contributor to the progression of tissue damage after stroke[11]. They arise in a spontaneous fashion in the penumbra[22], where blood flow is compromised[38], spreading into all areas that can respond. SDs can be triggered by either episodic drops in collateral flow or metabolic supply (e.g., hypoxic or hypotensive transients)[39–41] or increased metabolic demand (e.g., functional activation, movement, or touch); both situations create supply/demand mismatch in susceptible peri-infarct hot zones[40]. Reduced tissue perfusion delays metabolic recovery[42], prolonging SDs in the penumbra. Infarct expansion correlates both with the number of SDs[43–45] and their cumulative duration[33]. SDs artificially evoked in the ischemic hemisphere can also accelerate infarct growth proportional to their number[46,47], and suppression of SDs can ameliorate injury[7,48]. Due to their impact on secondary stroke expansion it is crucial to characterize and eventually treat SDs in a phase-specific fashion during the evolution of stroke to decrease secondary infarct expansion (though not yet a clinical reality).

Our central hypothesis is that the functional evolution of stroke boundaries is dynamic and evolves over the course of several days, with the frequency and duration of spontaneous depolarizations over time as we as collaterals and cerebral blood flow, determining the fate of the penumbra. Our objective here was to create an in vivo approach to continuously map stroke evolution and salvageable penumbra over days in an awake animal. For this purpose we adapted a high-density μECoG array recording setup in our stroke studies, previously extensively developed and tested for long-term use[16–18] in rodents, to determine the effects of SDs on correlative functional stroke size in awake, freely moving rats after focal cerebral ischemia. A more complete understanding of stroke evolution and the contribution of depolarizations in the non-anesthetized brain to infarct expansion will provide a platform for developing advanced diagnostic and treatment strategies that are phase-specific and personalized to improve stroke outcomes in our patients by detecting and then inhibiting SDs formation and propagation.

SDs are electrophysiologically complex, dynamic, and typically spread across the brain with a slow rate[12], so it is important to distinguish them from potential artefacts, ambient noise or fluctuations in the baseline brain activity. The variable nature of SD waveform, morphology, and spread during ischemia (where SDs can originate in and travel through tissue of variable health) complicates their reliable recognition and detection compared to SDs in a completely healthy brain, which are more stereotyped and feature a classical and often uniform negative shape (when recorded intracortically, in reference to an indifferent ground). In untreated stoke, the ongoing ischemia leads to a continuous deterioration of the surrounding brain tissue (i.e., progressive core enlargement), eventually impacting the tissue's ability to repolarize and recover from each event, which is mirrored in the change of ECoG activity and patterns.

The importance of observing ECoG changes is twofold:

1. Recognition of SDs, apart from specific characteristics defining their waveform shape, duration and spread, can be further substantiated by their associated ECoG suppression, which follows the initial depolarization in tissue that still has viability. Loss of detectable ECoG activity that is maintained after SDs indicates progression of tissue ischemia.

2. ECoG activity is an important indicator of tissue status after injury and can indicate the progression of injury or amelioration of it. Direct electrocorticographic (ECoG) patterns are characteristic during stroke phases[31,49–51], but differentiation of ECoG parameters recorded from areas *representing the ischemic core and penumbra* in stroke has been limited due to low density of ECoG contacts. For example, permanent middle cerebral artery occlusion in rats and humans leads to abnormal ECoG patterns. While the parietal region (peri-infarct) is dominated by sharp waves and/or spike-wave complexes, the ECoG in temporal regions (ischemic core) is severely depressed at 24 hours after MCAO[50]. Also, polymorphic delta activity has been reported predominantly in the peri-infarct parietal region and less in the core temporal region after transient MCAO[49]. However, these data were obtained during approaches limited to only 1–5 silver screws/hemisphere without the capability to detect SDs directly.

Our high-density data acquisition system for neural signal acquisition[17] has near DC-coupled amplifiers and gold contacts electroplated with PtIr[16,17], to lower the impedance of the array. Typically, the electrode impedance of standard gold or platinum metal electrodes is high at low frequencies (such as the frequency of SDs), making recordings more susceptible to noise. PtIr electrode coatings dramatically reduced the electrode impedance at low frequencies, enabling reliable recordings of SDs but, as with any metal electrode, potential waveform changes from asymmetric mobility (compared to ionic recordings). Thin polymer encapsulation layers produced a thin (<50 μm) and flexible electrode array. This configuration allowed us to place the array under the skull while leaving the dura mater intact, which maintains physiological CSF flow and does not impair normal brain physiology. The thin array allows us for the first time, to the best of our knowledge, to record SDs across a broad cortical area without a major craniotomy, since it can easily slide under the skull and above the dura through a narrow slit.

We have confirmed the feasibility of recording for days from awake animals (as previously demonstrated[16–18]), but also have critically demonstrated acute direct comparisons to glass penetrating electrodes. The events recorded in both are very similar, but the cortical arrays more commonly have an overshoot after the SD occurrence, due to potentially the extra-cortical location, metal vs ionic recordings, and filtering. This may be due to a combination of surface rather than intracortical recordings, different filtering (i.e., near DC vs DC) and the larger area of recording of the array contacts (compared to the very small glass electrodes). Additionally, the recording differences may be due to contamination from other ongoing events (such as cerebral blood flow changes), which are also recordable from metal electrodes, as well as signal contamination from the combined skull screw reference. Since we can record at near-DC frequencies, we can review the full raw signal without applying any filters, which allows clarifying potential filter-dependent distortion of the signal. This seems to be most

important for long-lasting SDs, where the filtering forces the signal back to baseline, whereas the raw signal (as demonstrated in Fig. 5) displays more drift but also a more accurate duration of the event.

We have demonstrated low electrode impedance and high SNR, allowing us to reliably detect SDs in a dynamic environment. Polarizable metals are commonly used for implantable electrodes due to biocompatibility concerns, particularly PtIr. Such materials present two concerns for recording very slow shifts in neural potential, as detailed in Hartings, et al[27]. The capacitive characteristic of these metals causes the electrode impedance to be roughly inversely proportional to frequency (an ideal polarizable material would completely block DC potential). However, the impact of electrode impedance on the amplified potential is dependent on the input impedance of the amplifier, which affects the proportion of voltage change across the electrode[52]. The impedance of our PtIr plated electrodes was 600-700 kΩ at 0.05 Hz, and the front-end amplifiers (TI OPA2376) had an input impedance of 1 GΩ. These impedance ratios suggest that the attenuation of cortical potential magnitudes due to the electrode was < 0.1% at frequencies ≥ 0.05 Hz. The second and more serious concern, is the spontaneous DC shift of several mV caused by slow drift of electrode polarization, accentuated by animal motion, which can eventually saturate the amplifiers. Using relatively low gain, the headstage amplifiers did not saturate during periods up to 132 hours of continuous recording. We attempted to measure the ambient potential drift by calculating the linear trend on every electrode for two minutes of signal sampled every hour. The majority of the drift (80%) was between -0.89 and 0.86 mV/min, which was small compared to faster rate of SDs. 98% of the drift was between -3.59 and 2.30 mV/min, but, upon inspection, many of the larger drift rates were from segments including the initiation or recovery of SDs and were not representative of true ambient drift. Since all human studies also use PtIr metal electrode contacts (and often a metal ground wire) these same concerns about polarization, asymmetric mobility (compared to ionic solutions) and mis-matched, unbalanced grounding (i.e., metal screws vs Ag-AgCl) can all lead to waveform distortion, in addition to the extracortical electrode location, requiring more complex rules of SD identification.

During our acute experiments with concurrent array and intra-cortical electrodes we sometimes failed to detect very small SDs (<1 mV). This might be due to the epidural location of our arrays, which in general decreases the overall amplitude of SDs compared to intra-cortical electrodes and can change waveform morphology. We further find it difficult to reliably confirm highly localized SDs, i.e. SDs that may involve only one or two channels. This limitation could only be resolved through simultaneously recording from more invasive penetrating electrodes, but which would require an invasive approach.

We also had difficulty in discerning single SDs when they appeared within a cluster (i.e. rapid consecutive SDs with no full repolarization). While we can reliably account for the total duration of a cluster, we may not be able to distinguish the exact beginning of an SD versus the end of a previous SD within a cluster. We can, however, determine if a cluster is a result of a circling SD using our topographic maps. While differentiation of overlapping SD events remains a limitation, the total duration of SDs also contributes to secondary infarct progression, which we can detect with high reliability.

In summary, we have developed and extensively characterized a recording system useful in awake animals that will allow us to define long-term stroke characteristics in the non-anesthetized brain, delineating post-injury phases, SD triggers, and chronic stroke evolution. Our findings could have a profound effect on patient care in the ICU setting if a suitable translational approach could be defined for less invasive array placement (i.e., than a typical craniotomy). Once reliable SD events and ECoG can be recorded over large brain areas, these recordings would provide a platform for developing advanced treatment strategies that are phase-specific and personalized to improve stroke outcomes, by inhibiting SD formation and propagation, and defining dynamic stroke evolution.

## Data availability

Data are available at request from the corresponding authors. Numerical source data for all figures can be found here: https://doi.org/10.6084/m9.figshare.25008488[53].

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

## Acknowledgements

Research reported in this publication was supported by the National Institute of Neurological Disorders and Stroke of the National Institutes of Health under Award Number R21NS111093 (to UH). The content is solely the responsibility of the authors and does not necessarily represent the official views of the National Institutes of Health. Kay Palopoli-Trojani received support from the E. Bayard Halsted Scholarship in Science, History and Journalism from The Graduate School at Duke University.

## Author contributions

K.P., M.T., D.A.T. and U.H. designed experiments, conducted experiments, and analyzed data, C.C., C.W., A.J.W. and J.V. designed the hardware and helped with data processing, C.L.E. performed experiments, K.P., M.T., C.C., D.A.T. and U.H. wrote the manuscript.

## Competing interests

C.C., C.W. and J.V. are inventors on US patent application US17/761,369, "Electroencephalography (eeg) electrode arrays and related methods of use," which covers aspects of the technology described here. All other authors declare no competing interest.
