## [Peer Review File · Communications Biology]

Reviewers' comments:

Reviewer #1 (Remarks to the Author):

The paper by Palopoli-Trojani et al., titled "High-density cortical μ ECoG arrays concurrently track peri-infarct depolarizations and long-term evolution of stroke in awake rats" describes a useful methodological advance in the design of large surface electrode recording arrays with low enough impedance contacts to permit near-DC recording of spreading depolarizations (SD) and peri-infarct field depolarizations (PID) for long periods of time in awake, freely-moving animals.

While the method has potential for translation to clinical monitoring of the frequency and, up to a point, spread of SDs and PIDs following stroke and other brain injuries, there are limitations that are acknowledged by the authors, but not sufficiently evaluated for the magnitude of their effect over the course of their long-term recordings.

Specifically:

1) The data illustrate clearly the problems with inferring the extend of abnormal penumbral tissue from even large numbers of extracellular recording contacts over the surface of the brain. The two figures that show correlated depth electrode recordings (Figs. 4 & 5) show that the patterns of potentials seen at various array contacts are very complex and quite different from the depth recording. In extracellular recordings, there are many factors that can make multiple, differing types of neural events result in the same EEG pattern. It is not going to be that easy to interpret the borders of penumbral tissue from the SD and PID patterns picked up by the array surface electrodes. I would have more confidence that the evolution of this penumbral region over time, if the authors addressed number 2 below:

2) While the authors do state they were able to record for up to 132 hours without DC drift saturating their amplifiers, there needs to be more quantitative evaluation of the pattern and magnitude of changes in recordings over long time periods, and the stability of patterns from these arrays over time, to know how much of the changes are stroke-induced. This could be done by evoking, in normal brain, PIDs and SDs, and determining whether the activity pattern recorded changes over the 132 hours or longer, and how this correlates with impedances changes in the array contacts, etc.

3) Any surface array will preferentially pick up activity from large numbers of neurons with stereotyped circuitry anatomy closer to the recording contacts. Thus, deep brain strokes will be much harder to evaluate than surface cortices, and the method will be most useful for human strokes affecting primarily cerebral cortices, provided less invasive surgical methods for inserting the flexible array can be used safely. Since large array surface EEG contacts can also pick up such SD activity, the authors should address to what extent their methods may offer advantages that outweigh the necessity for invasive insertion of the array over the dura.

4) While the goal of this methodological advance is admirable, it would be greatly strengthened by some independent measure of the evolution of the infarct penumbra over time, to allow validation of the sensitivity and reliability of their recording array in monitoring spread of ischemic damage-related SDs and eventual irreversible tissue damage over time.

Overall, this is a methodological advance that will only be able to influence the field to the extent that it can be replicated by other investigators, rigorously evaluated for its reliability and resolution in being able to monitor penumbral spread of damage over time, and eventual translation to the clinic. The current report needs more evaluation of stability of recordings over time, correlation of these

patterns with depth recordings, indication of advantages it offers over multi-contact EEG, and how well it can predict underlying tissue damage evolution over time.

Reviewer #2 (Remarks to the Author):

In this very interesting manuscript, the authors present an electrode array with 61 contacts suitable for long term recording of spreading depolarizations and physiological brain activity with high spatial resolution. They nicely approached the problem of long-term DC drifts by seeking good balance between signal amplification and ADC-resolution. They tested the system comparing array recordings of potassium and stroke induced spreading depolarizations with recordings obtained from interstitial microelectrode.

In my opinion this recording system could also be used in pathological condition distinct from stroke for exactly the same purpose as described in the manuscript – recording and characterization of deleterious spreading depolarizations and eventually for monitoring possible therapeutic approaches.

This approach seems to be very useful, nevertheless I see some issues with the manuscript:

1. Terminology: Although the term periinfarct depolarization was used for many years in the stroke research, since the publication of the recommendation of the COoperative Studies on Brain Injury Depolarizations (COSBID) (Dreier et al., 2017) the more generic term “spreading depolarization” is more appropriate. It describes better the continuum ranging from SD occurring in migraine aura to the terminal spreading depolarization after cardiac arrest and is not restricted to any particular disease. There is most likely no substantial difference between SDs occurring in the penumbra of ischemic infarct and in the traumatic lesion as both contribute to the lesion growth, but one cannot really use the term “PID” in the second condition.

2. Frequency bands: The reason why the authors separated the analyzed frequency bands at 2 Hz is not clear. According to clinical EEG textbooks the physiological activity (delta band) starts at 0.5 Hz. These frequencies between 0.5 and 2 Hz may substantially contribute to the amplitude of the signal. Thus, analysis of spreading depression should incorporate frequencies from 0.5 Hz onward. The depolarization wave is usually recorded in frequencies below 0.05 Hz (Dreier et al., 2017) and this does not interfere with frequency bands important for neuronal activity. Therefore, I suggest reanalyzing the recording with the appropriate split frequency.

3. Long-term ECoG recording: In this manuscript authors use a small craniotomy and apparently slide the array between the dura and the bone. In the previous publications cited here (page 5, line 129, ref. 20 and 45) always a craniotomy with 5x5 mm (bigger than the array) was performed. What is the impact of this new technique on the brain? How frequently bleeding from dural vessels occurred? Were there cases of epidural hematoma? Even if these questions are not crucial for this manuscript, they have to be answered before the authors start any preclinical study (as they apparently plan to proceed).

4. ECoG-Measurements in stroke patients (page 4, lines 104-106): There is indeed clear indication for decompressive hemicraniectomy in stroke patients – the so called malignant hemispheric stroke (Vahedi et al., 2007) which is included in multiple guidelines for stroke treatment. Therefore there is no lack of rationale for surgery.

In operated stroke patients the electrode strip is usually placed across the infarct border covering as well the infarct core as the part of the penumbra (Dohmen et al., 2008; ref 44). Why do the authors think that this recording is “skewed” compared to their approach?

5. Positive DC-shifts in epidural recordings: When measured epidurally, spreading depolarization can be followed by positive DC shift (Kang et al., 2013), so the first three depolarizations shown in fig. 5b are most likely represented by the negative deflections of the signal preceding the positive ones.

6. Isoelectric SDs: The inner part of the penumbra with low perfusion can lead to neuronal hyperpolarization, which inhibits the generation of action potentials (so called non spreading depression). These cells still can depolarize, which then results in isoelectric spreading depolarizations (Dreier et al., 2017). Thus, spreading depolarization in the penumbra not always induce depression of physiological neuronal activity. This might also explain the difference in the penumbra border between depression and depolarization (line 447). Did the authors observe isoelectric SDs in their recording?

7. Negative ultraslow potential (NUP): The clusters of SDs without repolarization mentioned on page 30, line 689 might be a part of a NUP which is a sign of infarct growth/development (Lückl et al., 2018) and might be worth of additional analysis.

Minor remarks:

- Page 3, line 74-75: In the clinical setting core and penumbra are usually identified by diffusion/perfusion mismatch in the MRI shortly after the infarct occurred and not after days.

- Page 6, line 174: Is reference 21 the same as 20? (40 no details on OP but subdural)

- Page 5, line 139: Most likely "spreading depolarization" should be used instead "spreading depression" as the first one is detected in low frequency range.

- Page 25, line 549: SDs cannot invade the core because the cells inside are already permanently depolarized (compare ref. 18) This paragraph should be reformulated for clearance.

- Fig 7: The depolarizations seem to occur at the same time in all channels without any spread. According to ref. 11 spread between different channels is less recognizable in epidural than in subdural recordings. Nevertheless, the examples in submitted movies seem to show more clear time delay between distant electrodes. An example with clear spread of depolarization and depression would be more illustrative in this case (if available).

References:

Dohmen, C. et al. (2008) 'Spreading depolarizations occur in human ischemic stroke with high incidence', *Ann Neurol*, 63(6), pp. 720–8. doi: 10.1002/ana.21390.

Dreier, J. P. et al. (2017) 'Recording, analysis, and interpretation of spreading depolarizations in neurointensive care: Review and recommendations of the COSBID research group', *Journal of Cerebral Blood Flow & Metabolism*, 37(5), pp. 1595–1625. doi: 10.1177/0271678X16654496.

Kang, E. J. et al. (2013) 'Blood–brain barrier opening to large molecules does not imply blood–brain barrier opening to small ions', *Neurobiology of Disease*, 52, pp. 204–218. doi: 10.1016/j.nbd.2012.12.007.

Lückl, J. et al. (2018) 'The negative ultraslow potential, electrophysiological correlate of infarction in the human cortex', *Brain: A Journal of Neurology*. doi: 10.1093/brain/awy102.

Vahedi, K. et al. (2007) 'Early decompressive surgery in malignant infarction of the middle cerebral

artery: a pooled analysis of three randomised controlled trials', *The Lancet Neurology*, 6(3), pp. 215–222. doi: 10.1016/S1474-4422(07)70036-4.

Reviewer #3 (Remarks to the Author):

The authors present high density ECoG arrays for chronic recordings of DC and high frequency activity over large cortical areas in non-anesthetized and non-restrained rodents following MCAO. This technology has important advantages over existing techniques and the results sound very interesting. However, there are several points that should be considered.

1) My main concern is that this paper lacks quantifications, statistical descriptions and comparisons. Most notably, should be provided, at the group level, and compared between intracortical and ECoG electrodes: amplitude and temporal parameters of SDs and PIDs, SNR values for SD and PIDs, it should be explained how was set threshold for PIDs detection, and how their amplitude, onsets and offsets were calculated (e.g. at the example traces shown on Fig. 5b PIDs are actually hardly detectable by eye), how the borders of penumbra were defined on Fig. 5-7 and how these were related to histological damage assessed with TTC staining; characterize the dynamics of penumbra progression, and whether this was specifically associated with prolongation of PIDs / irreversible DC shifts

2) Fig 3 : it would be nice showing here example traces during natural slow-wave sleep. Top blue recordings are probably from awake animals, and "awake" should be instead of "baseline". Also, it would be nice showing examples of delta-wave and isoflurane-induced population burst propagation through the array at expanded time scale.

3) Fig. 4: please indicate depth of intracortical recordings

4) KCl-induced SDs are described in the text, but these should also be presented on figures together with quantitative comparisons at the epidural grids and intracortical recordings.

5) Fig 5: a) how penumbra borders were identified? How this image relates to the TTC-stain? b) indicate PIDs detected on intracortical and ECoG electrodes; c-d) provide cross-correlation between data shown on c and d

6) Fig. 7: Please provide quantification of spatial progression of ischemia relying on electrophysiological criteria

July 12th, 2023

Re: Decision on manuscript COMMSBIO-20-0414-T

We sincerely appreciate the opportunity to respond to the review of our paper and are happy to address the questions from the reviewers.

Reviewer #1 (Remarks to the Author):

The paper by Palopoli-Trojani et al., titled “High-density cortical μ ECoG arrays concurrently track peri-infarct depolarizations and long-term evolution of stroke in awake rats” describes a useful methodological advance in the design of large surface electrode recording arrays with low enough impedance contacts to permit near-DC recording of spreading depolarizations (SD) and peri-infarct field depolarizations (PID) for long periods of time in awake, freely-moving animals.

While the method has potential for translation to clinical monitoring of the frequency and, up to a point, spread of SDs and PIDs following stroke and other brain injuries, there are limitations that are acknowledged by the authors, but not sufficiently evaluated for the magnitude of their effect over the course of their long-term recordings.

Specifically:

1) The data illustrate clearly the problems with inferring the extend of abnormal penumbral tissue from even large numbers of extracellular recording contacts over the surface of the brain. The two figures that show correlated depth electrode recordings (Figs. 4 & 5) show that the patterns of potentials seen at various array contacts are very complex and quite different from the depth recording. In extracellular recordings, there are many factors that can make multiple, differing types of neural events result in the same EEG pattern. It is not going to be that easy to interpret the borders of penumbral tissue from the SD and PID patterns picked up by the array surface electrodes. I would have more confidence that the evolution of this penumbral region over time, if the authors addressed number 2 below:

Thank you for this important comment. We agree; recordings from the depth electrode are much clearer to interpret (although only in one small spot). However, any less invasive extracellular recording will have to find a way to overcome the difficulty you describe. Therefore, our approach offers an advantage: we can not only discern the penumbra with regard to specific SDs and PID patterns; with the given density of contacts in the array, we have multiple detection points, which allows to a) confirm spread, b) track where we see pattern change and c) most importantly: we can concurrently use the cortical (ECoG) activity to derive the penumbra borders.

2) While the authors do state they were able to record for up to 132 hours without DC drift saturating their amplifiers, there needs to be more quantitative evaluation of the pattern and magnitude of changes in recordings over long time periods, and the stability of patterns from these arrays over time, to know how much of the changes are stroke-induced. This could be done by evoking, in normal brain, PIDs and SDs, and determining whether the activity pattern recorded changes over the 132 hours or longer, and how this correlates with impedances changes in the array contacts, etc.

We thank the reviewer for this question. Prior work, shown by our group, has shown that the impedance of five implanted arrays remained stable for 247–435 days in vivo, and accelerated aging predicts device integrity beyond 3.4 years (Woods et al. 2018), which is well beyond the length of our recordings in MCAO rats (which are at most 10 days). Further, we have quantified the DC levels and drift rate of our recording system in the figure below. The DC levels of our recording system remain stable, and the fluctuations we do observe are 1. of much smaller magnitude compared to the amplitude of PIDs we record and 2. do not have the same dynamics/morphology as PIDs. Given these two considerations, we can confidently identify stroke-induced PIDs without interference from the recording system's DC drift. We included the figure below in the main manuscript, as supplemental figure 2.

3) Any surface array will preferentially pick up activity from large numbers of neurons with stereotyped circuitry anatomy closer to the recording contacts. Thus, deep brain strokes will be much harder to evaluate than surface cortices, and the method will be most useful for human strokes affecting primarily cerebral cortices, provided less invasive surgical methods for inserting the flexible array can be used safely. Since large array surface EEG contacts can also pick up such SD activity, the authors should address to what extent their methods may offer advantages that outweigh the necessity for invasive insertion of the array over the dura.

Thank you for this very important comment. Please allow us to address the inherent suggestions:

1. We agree fully that our method reported here remains restricted to the cortex. With potential future translation in mind, the goals of our monitoring method are to remain as little invasive as possible and yet identify SDs (and penumbra changes) reliably covering a sufficient surface area. This is important, as ischemia and injury sites grow over time. Eventually the goal is to prevent secondary lesion growth: to document either change, we need an “electrode net” to discern dynamic change. Insertion of many in depth electrodes as is possible in invasive Seizure mapping, would be more distinct but also more invasive and particularly within injured cortex (potentially causing hemorrhage) the risks in the current opinion are too high. We would like to state though, that cortical spreading depolarization currently are believed to be occurring almost exclusively in the cortex, although first data have been reported that SD spread may occur into deeper brain structures (Eikermann et al).
2. Non-invasive surface EEG contacts would be the ideal way to measure SDs after brain injury, however this has not been achieved yet and faces many complex problems as described by Hofmeijer et al. (2018) [2]. Hofmeijer and colleagues used scalp EEG to monitor 18 patients with ischemic stroke and 18 patients with TBI, but they were unable to identify SDs unambiguously using EEG.
3. In humans, SD monitoring has been developed by the COSBID group, using subdural ECoG strips [1], requiring invasive craniotomies in patients that developed malignant strokes or had other indications for decompressive craniotomies.
4. In experimental animals, especially rodents, surface EEG recordings of SDs have not been reliably possible, so we still rely on bypassing the capacitance of the skull, placing the arrays epidurally, which is already an advancement, since we do not need to open the dura.

4) While the goal of this methodological advance is admirable, it would be greatly strengthened by some independent measure of the evolution of the infarct penumbra over time, to allow validation of the sensitivity and reliability of their recording array in monitoring spread of ischemic damage-related SDs and eventual irreversible tissue damage over time.

Thank you for this very insightful comment, as it addresses exactly the underlying problem that led to the development of our approach. Currently, it is challenging to monitor the infarct penumbra over time in awake, freely behaving animals. We are not aware of any current approach that could achieve this goal, other than functional assessment of the ECoG signal. Repetitive imaging (several times within 24 hours) would probably give some insight into this process, but would at

the very least (in animals) require repetitive sedation, which in stroke animals would massively increase mortality and certainly confound stroke progression, and therefore the study. It is further practically not doable. Histologic assessment would always require the sacrifice of the animal, which would require enormous animals numbers. Eventually this approach would be justified, in our opinion, once we had a functional outcome measure method established, here our continuous ECoG and PID assessment. To be able to track the penumbra over time in non-anesthetized animals is one of the most challenging and yet most needed tools for the future development of neuroprotective strategies in stroke. We are presenting here step 1, which monitors functional parameters.

Overall, this is a methodological advance that will only be able to influence the field to the extent that it can be replicated by other investigators, rigorously evaluated for its reliability and resolution in being able to monitor penumbral spread of damage over time, and eventual translation to the clinic. The current report needs more evaluation of stability of recordings over time, correlation of these patterns with depth recordings, indication of advantages it offers over multi-contact EEG, and how well it can predict underlying tissue damage evolution over time.

We agree with the reviewer. We have attached the drift rate over time (supplemental figure 2) to address the question about stability of recordings over time. Correlations of depth recordings over time in awake moving animals may be an ideal situation, but seems an unrealistic ask for this setting. Multi-contact EEG again in freely moving animals through a thin skull may allow EEG activity to be detected, but at this moment in time does not allow concurrent detection of spreading depolarizations, as the skull (also the most significant barrier in humans) will function as a filter that makes reliable SD detection, in particular with regard to moving artefacts in moving animals, almost impossible.

We are aware that our method has limitations, however it can provide new insight into the secondary infarct progression in awake moving animals over days, which is a substantial departure from the current status quo.

Lastly, previous work has been published that justifies sub-millimeter sampling of cortical neural signals obtained with μ ECoG arrays [3]. Though this study doesn't specifically talk about low frequency activity like SDs, it suggests that high resolution ECoG recordings would be advantageous compared to low density EEG when monitoring spontaneous neural activity to delineate core and penumbra borders. ECoG arrays are already widely used by investigators all around the world for other applications: our study demonstrates the advantages of using this technology to study SDs over multi-contact EEG.

Reviewer #2 (Remarks to the Author):

In this very interesting manuscript, the authors present an electrode array with 61 contacts suitable for long term recording of spreading depolarizations and physiological brain activity with high special resolution. They nicely approached the problem of long-term DC drifts by seeking good balance between signal amplification and ADC-resolution. They tested the system comparing array recordings of potassium and stroke induced spreading depolarizations with recordings obtained from interstitial microelectrode.

In my opinion this recording system could also be used in pathological condition distinct from stroke for exactly the same purpose as described in the manuscript – recording and characterization of deleterious spreading depolarizations and eventually for monitoring possible therapeutic approaches.

This approach seems to be very useful, nevertheless I see some issues with the manuscript:

1. Terminology: Although the term periinfarct depolarization was used for many years in the stroke research, since the publication of the recommendation of the COoperative Studies on Brain Injury Depolarizations (COSBID) (Dreier et al., 2017) the more generic term “spreading depolarization” is more appropriate. It describes better the continuum ranging from SD occurring in migraine aura to the terminal spreading depolarization after cardiac arrest and is not restricted to any particular disease. There is most likely no substantial difference between SDs occurring in the penumbra of ischemic infarct and in the traumatic lesion as both contribute to the lesion growth, but one cannot really use the term “PID” in the second condition.

Thank you for this comment. We have now replaced the term peri-infarct depolarization (PID) with spreading depolarization (SD) for broader applicability also in other acute brain injuries.

2. Frequency bands: The reason why the authors separated the analyzed frequency bands at 2 Hz is not clear. According to clinical EEG textbooks the physiological activity (delta band) starts at 0.5 Hz. These frequencies between 0.5 and 2 Hz may substantially contribute to the amplitude of the signal. Thus, analysis of spreading depression should incorporate frequencies from 0.5 Hz onward. The depolarization wave is usually recorded in frequencies below 0.05 Hz (Dreier et al., 2017) and this does not interfere with frequency bands important for neuronal activity. Therefore, I suggest reanalyzing the recording with the appropriate split frequency.

Thank you for the insightful comment. We have re-analyzed our data in the suggested frequency bands. We have updated the figures (2, 5, and 7) to reflect the new frequency band analysis. The overall results remained the same.

3. Long-term ECoG recording: In this manuscript authors use a small craniotomy and apparently slide the array between the dura and the bone. In the previous publications cited here (page 5, line 129, ref. 20 and 45) always a craniotomy with 5x5 mm (bigger than the array) was performed. What is the impact of this new technique on the brain? How frequently bleeding from dural vessels occurred? Were there cases of epidural hematoma? Even if these questions are not crucial for this manuscript, they have to be answered before the authors start any preclinical study (as they apparently plan to proceed).

Thank you for this question. We have edited the method section for clarity. For chronic awake recordings, we have performed a small slit craniotomy, without any further injury, leaving the dura intact (2mm width and 5mm length) to insert the array under the bone.

This must be distinguished from the acute non-survival experiments under anesthesia, which we performed to achieve simultaneous recordings with the intra-parenchymal electrodes. In order to place both the array, KCL application side and intra-parenchymal electrode, as well as ground electrodes, we needed to make a larger craniotomy i.e. 5x5 mm.

In our chronic recording experiments, the experimental protocol included a routine autopsy of the animal after the end of the experiment. We have assessed the presence of epidural hematomas, excessive scar formation and infection.

After our initial pilot experiments, we observed one animal that had a small epidural hematoma present at the end of the recording time, which was clinically unapparent and did not influence the recording quality. However, data from this animal was not included in this manuscript. We did not find evidence for infection or scar formation in our animals.

4. ECoG-Measurements in stroke patients (page 4, lines 104-106): There is indeed clear indication for decompressive hemicraniectomy in stroke patients – the so called malignant hemispheric stroke (Vahedi et al., 2007) which is included in multiple guidelines for stroke treatment. Therefore there is no lack of rationale for surgery. In operated stroke patients the electrode strip is usually placed across the infarct border covering as well the infarct core as the part of the penumbra (Dohmen et al., 2008; ref 44). Why do the authors think that this recording is “skewed” compared to their approach?

Thank you for this question. We have edited the sentence for clarity in the manuscript. We aimed to address the problem that ECoG measurements in patients are exclusively performed in malignant hemispheric stroke (or severe head injury) patients, that require decompressive surgery to alleviate the increase in intracranial pressure. Therefore, our current knowledge on SD occurrence is skewed towards very severe strokes/severe head injury. It is unclear if more focal or less severe strokes elicit less frequent SDs, if SD occurrence is limited to shorter post-ischemic time periods, if SDs occur in clusters, to name only a few unknowns. We believe our current information is skewed towards the most severe stroke cases and do not represent most strokes, that remain unmonitored.

5. Positive DC-shifts in epidural recordings: When measured epidurally, spreading depolarization can be followed by positive DC shift (Kang et al., 2013), so the first three depolarizations shown in fig. 5b are most likely represented by the negative deflections of the signal preceding the positive ones.

Thank you for this reference, we have included it in the manuscript. The stereotyped SD shape we see in the array recordings does indeed include a positive DC shift following initial depolarization.

6. Isoelectric SDs: The inner part of the penumbra with low perfusion can lead to neuronal hyperpolarization, which inhibits the generation of action potentials (so called non spreading depression). These cells still can depolarize, which then results in isoelectric spreading depolarizations (Dreier et al., 2017). Thus, spreading depolarization in the penumbra not always induce depression of physiological neuronal activity. This might also explain the difference in the penumbra border between depression and depolarization (line 447). Did the authors observe isoelectric SDs in their recording?

We have edited Figure 5 panels c and d (line 447). The original figure showed RMS voltage in two different frequency ranges (0.005-2 Hz and 2-200 Hz), where the low frequency range was analogous to SD amplitude. We performed further analyses and concluded that spatial distributions (heatmaps) of SD amplitude are not very informative. The reason is because each channel will record a different amplitude for the same SD (based on each channel's proximity to the SD at each time point). Therefore, the heatmaps of SD amplitude are not informative because they suggest that each channel is recording a different SD, when in fact each channel is recording the same SD event. We thus repeated the analysis to show RMS voltage in one frequency range (greater than 0.5 Hz, the frequency range of ECoG that is suppressed during SDs, as Reviewer #1 indicated) before and during SDs to show spatial suppression of activity during SDs, which is of more importance from a pathophysiology perspective of stroke progression.

7. Negative ultraslow potential (NUP): The clusters of SDs without repolarization mentioned on page 30, line 689 might be a part of a NUP which is a sign of infarct growth/development (Lüeckel et al., 2018) and might be worth of additional analysis.

Thank you for this comment. We are currently conducting more long term stroke experiments and are analyzing the data sets. We look forward to discussing NUPs in a future manuscript, if observed.

Minor remarks:

- Page 3, line 74-75: In the clinical setting core and penumbra are usually identified by diffusion/perfusion mismatch in the MRI shortly after the infarct occurred and not after days.

Thank you for this comment. The window to identify diffusion/perfusion mismatch closes very soon after the initial stroke onset, which makes it difficult or even impossible to delineate the

penumbra over the first several days after stroke through any conventional criteria, including blood flow, imaging (clinical MRI, CT, direct optical imaging), and angiography. We have edited the sentence.

- Page 6, line 174: Is reference 21 the same as 20? (40 no details on OP but subdural)

References 20 and 21 are the same citations. Thank you for pointing out this mistake, we have removed the duplicate citation.

- Page 5, line 139: Most likely “spreading depolarization” should be used instead “spreading depression” as the first one is detected in low frequency range.

We have corrected the term throughout the manuscript.

- Page 25, line 549: SDs cannot invade the core because the cells inside are already permanently depolarized (compare ref. 18) This paragraph should be reformulated for clearance.

We have edited the sentence for clarity.

- Fig 7: The depolarizations seem to occur at the same time in all channels without any spread. According to ref. 11 spread between different channels is less recognizable in epidural than in subdural recordings. Nevertheless, the examples in submitted movies seem to show more clear time delay between distant electrodes. An example with clear spread of depolarization and depression would be more illustrative in this case (if available).

The example shown in Figure 7 was chosen to highlight a cluster of SDs that has increased suppression of ECoG during each subsequent SD. We chose this example to highlight the ability of our arrays to detect different levels of ECoG suppression. Increased duration of SDs and SD clusters is indicative of worsening tissue conditions post stroke, and therefore it is extremely crucial to be able to evaluate the effect of longer duration of PIDs and clusters of SDs on ECoG suppression.

Limitation: one of the limitations of the method is that we cannot determine objective amplitude sizes, as we have to take into account ground and so forth, tissue status, electrode position, so our focus is on reliably detecting SDs and their duration, and ECoG suppression as indicator for the general tissues status, not so much quantification of SD amplitude like in intraparenchymal electrodes.

References:

Dohmen, C. et al. (2008) ‘Spreading depolarizations occur in human ischemic stroke with high incidence’, *Ann Neurol*, 63(6), pp. 720–8. doi: 10.1002/ana.21390.

Dreier, J. P. et al. (2017) 'Recording, analysis, and interpretation of spreading depolarizations in neurointensive care: Review and recommendations of the COSBID research group', *Journal of Cerebral Blood Flow & Metabolism*, 37(5), pp. 1595–1625. doi: 10.1177/0271678X16654496.

Kang, E. J. et al. (2013) 'Blood–brain barrier opening to large molecules does not imply blood–brain barrier opening to small ions', *Neurobiology of Disease*, 52, pp. 204–218. doi: 10.1016/j.nbd.2012.12.007.

Lückl, J. et al. (2018) 'The negative ultraslow potential, electrophysiological correlate of infarction in the human cortex', *Brain: A Journal of Neurology*. doi: 10.1093/brain/awy102.

Vahedi, K. et al. (2007) 'Early decompressive surgery in malignant infarction of the middle cerebral artery: a pooled analysis of three randomised controlled trials', *The Lancet Neurology*, 6(3), pp. 215–222. doi: 10.1016/S1474-4422(07)70036-4.

Reviewer #3 (Remarks to the Author):

The authors present high density ECoG arrays for chronic recordings of DC and high frequency activity over large cortical areas in non-anesthetized and non-restrained rodents following MCAO. This technology has important advantages over existing techniques and the results sound very interesting. However, there are several points that should be considered.

1) My main concern is that this paper lacks quantifications, statistical descriptions and comparisons. Most notably, should be provided, at the group level, and compared between intracortical and ECoG electrodes: amplitude and temporal parameters of SDs and PIDs, SNR values for SD and PIDs, it should be explained how was set threshold for PIDs detection, and how their amplitude, onsets and offsets were calculated (e.g. at the example traces shown on Fig. 5b PIDs are actually hardly detectable by eye), how the borders of penumbra were defined on Fig. 5-7 and how these were related to histological damage assessed with TTC staining; characterize the dynamics of penumbra progression, and whether this was specifically associated with prolongation of PIDs / irreversible DC shifts

We thank the reviewer for their insightful comment. We have added statistics of the different SDs as reproduced in the table below.

Comparison of the amplitude and temporal patterns of KCl-induced SDs and post-MCAO PIDs, recorded from both an intracortical electrode and our arrays, in acute and chronic settings:

	Intracortical		Acute array		Chronic array	
	Amplitude (mV)	Duration (sec)	Amplitude (mV)	Duration (sec)	Amplitude (mV)	Duration (sec)
KCL SDs	21.4 ± 3.2 n=22	26.0 ± 5.6 n=22	16.7 ± 7.5 n=14	47.1 ± 9.1 n=14	n/a	n/a
Stroke SDs	18.9 ± 8.1 n=22	60.3 ± 35.0 n=22	9.8 ± 6.5 n=17	82.4 ± 42.5 n=17	3.2 ± 2.7 n=36	59.0 ± 28.2 n=36

To compare in vivo intracortical and array measurements of KCL elicited SDs and spontaneous occurring PIDs, it is critical to take the experimental setting into account in which such measurements are obtained. Principal demonstrated in figure 5, panel B.

1. KCL elicited SDs are obtained in otherwise healthy brains, with no underlying ischemia, so their occurrence is regularly imitated after the KCL if applied to the cortex. Intracortical SDs are measured with one 200 micrometers deep inserted glass micropipette, detecting

the summation potential at this one given point referenced to one silver-silver chloride ground electrode in the neck of the animal. Amplitudes have been measured from the onset of the mass DC depolarization to the maximum negative deflection point. Duration has been measured at half maximal amplitude. Array recordings have been performed at the same time via skull window preparation of approximately 5x5 mm. The array was referenced via silver wires to the stereotaxic frame and ground as well as to the head screw as in the final preparation, which is crucial to explain the amplitude differences. Further, in this preparation, the array had not been cemented in as that would have made the parallel insertion of the intracortical electrode impossible.

2. For the spontaneous PID detection, a permanent ischemia (filament MCAO) had been performed on the animal before the start of the recordings, following the same experimental setting as mentioned above. PIDs by the current definition do not have a unique shape, amplitude or duration, in fact these parameters can be highly variable and therefore statistics comparing duration in different stroke animals is complex and would require many experiments to be meaningful.

Our focus here, in this method paper was to provide evidence that we can detect SDs and PIDs reliably and define inclusion and exclusion criteria. Since the arrays had many electrodes, that represent events in different areas of the brain, reflecting different tissue statuses, it further complicates meaningful statistics. For the requested parameters, we have chosen the array channel that displayed the largest depolarization, subsequent ECoG suppression and clear spread. However, in any given SD event, 60 contacts display very different aspects of this event, as stated above, meaning, SD Amplitude can vary substantially from stroke core areas to inner and outer penumbra.

3. Part of our current work and accumulating data analysis is to specifically study and address characteristics of SDs for 10 days and up to 30 days. Here we aimed to demonstrate that we can reliably detect SDs in different settings (KCL, healthy brain, stroke).
4. The onset of any depolarization was calculated from the start of the sharp negative DC deflection to the maximum deflection point. Offset, if applicable, is defined as return to baseline. Duration was calculated using the full width at half maximum (FWHM). We would like to point out that correct recognition of SDs in the array recording is more complex and therefore we have defined step by step criteria to include or exclude an event.

2) Fig 3 : it would be nice showing here example traces during natural slow-wave sleep. Top blue recordings are probably from awake animals, and “awake” should be instead of “baseline”. Also, it would be nice showing examples of delta-wave and isoflurane-induced population burst propagation through the array at expanded time scale.

Thank you for your suggestion, we have edited the labels of Figure 3 accordingly.

3) Fig. 4: please indicate depth of intracortical recordings

Thank you, it is 200 micrometers. This information was added to the caption of Figure 4, as well as in the text under *Methods: Acute In Vivo Setup* and *Methods: Array Characterization In Vivo*.

4) KCl-induced SDs are described in the text, but these should also be presented on figures together with quantitative comparisons at the epidural grids and intracortical recordings.

We performed experiments with KCl induced SDs as a first step in validating our recording setup, since KCl induced SDs are much more stereotyped and therefore more appropriate for initial evaluation of our recording system. We have now provided a representative example of a KCl induced SD below, recorded from both our arrays and an intraparenchymal electrode, and have added this figure to the manuscript as Supplemental Figure 3. It is clear from the figure that our array is able to pick up the same SD as the penetrating electrode. Some channels on the array detect the SD at an earlier time point than the penetrating electrode; others detect the SD at a delayed time point. The broad cortical coverage of our array therefore allows us to view the temporal spread of the SD, which we otherwise cannot do with the penetrating electrode alone. Also important to note, the amplitude of the SD varies across channels of our array: this information can also be used to view the spatial spread of the SD. The smaller the amplitude of the SD at a given array channel, the more distant the SD is from the array (this could mean more distant as in more deeply located inside the brain, or more distant as in laterally on the cortex). This is beyond the scope of this paper, but the high density recordings of our array could be used in a source localization algorithm to recreate the exact origin of the SD (not only in the xy plane of the cortex, but also in the z direction, deep inside the brain).

**KCl induced spreading depolarization
Intraparenchymal electrode**

Staggered array channels

5) Fig 5: a) how penumbra borders were identified? How this image relates to the TTC-stain? b) indicate PIDs detected on intracortical and ECoG electrodes; c-d) provide cross-correlation between data shown on c and d

We edited Figure 5 panels c and d as we described in the response to Reviewer #2 Comment #6. a) The penumbra borders were identified by plotting the spatial heatmap of RMS voltage in the frequency range 0.5-200 Hz (updated Figure 5 panel c). This is the frequency range that is suppressed during PIDs. As tissue from the penumbra gets recruited into the core, it shows extremely suppressed ECoG activity (a direct result of the tissue no longer being viable). For this reason, we are able to draw penumbra borders by looking at the heatmap of ECoG power before or after PIDs (we can't look at cog power during PIDs to draw penumbra borders, because during PIDs there is transient suppression of ECoG power).

b) Unfortunately, we do not have a TTC stain for this experiment.

c) As we addressed in our response to Reviewer #2 Comment #6, heatmaps of SD amplitude (approximated by RMS power in the frequency range of SDs) are not informative. We therefore replaced panels c and d with heatmaps of RMS voltage in the frequency range 0.5-200 Hz (which is the range of ECoG that is suppressed during SDs) before and during SDs to show spatial suppression of activity during SDs, which is of more importance from a pathophysiology perspective of stroke progression. This, together with our response to Reviewer #3 Comment #1, is why we think that a correlation between panels c and d is not a meaningful statistical comparison: we have 60 recordings of the same event, which are slightly different from each other for a variety of reasons (as described in response to Reviewer #3 Comment #1). Moreover, in our updated Figure 5c, the ECoG power before PIDs (which is used to delineate the border of the penumbra) is indicative of the viability of the tissue - once a channel gets recruited into the core, it remains there. However, in our updated Figure 5d, the ECoG power during PIDs (which shows suppression of ECoG during PIDs) reveals a *transient* suppression. Therefore, it is not meaningful to perform a correlation between an *absolute* measure of tissue viability and a *transient* measure of suppression.

6) Fig. 7: Please provide quantification of spatial progression of ischemia relying on electrophysiological criteria

We have updated Figure 8 to show quantification of spatial progression of ischemia. We decided to show spatial progression of ischemia for Figure 8 (rather than Figure 7) because Figure 8 shows a full recording, from stroke induction until terminal depolarization and death. To show spatial progression of ischemia, we have shown the power spectrum of 3 different channels at different time points. One of the channels, the one closest to the core, shows extremely low power immediately after stroke, and the power remains that low throughout the recording. The other two channels show a decrease of power after multiple hours of SDs, indicating that they are being recruited into the penumbra as the stroke progresses.

References:

1. Dreier, J.P., et al., *Recording, analysis, and interpretation of spreading depolarizations in neurointensive care: Review and recommendations of the COSBID research group*. J Cereb Blood Flow Metab, 2017. **37**(5): p. 1595-1625.
2. Hofmeijer, J., et al., *Detecting Cortical Spreading Depolarization with Full Band Scalp Electroencephalography: An Illusion?* Front Neurol, 2018. **9**: p. 17.
3. Trumpis, M., et al., *Sufficient sampling for kriging prediction of cortical potential in rat, monkey, and human μ ECoG*. Journal of Neural Engineering, 2021. 18(3): p. 036011.

Reviewers' comments:

Reviewer #1 (Remarks to the Author):

The authors have supplied thoughtful responses to all of the reviews, and added additional analyses and supplemental data where useful.

This added content has significantly improved the utility of the work for future investigations.

I have satisfied with their additions and corrections, and have no further comments.

Reviewer #2 (Remarks to the Author):

I would like to acknowledge the substantial improvements made in the current version of the manuscript. Nevertheless, a few issues still require careful consideration.

1. Is a positive DC shift in an epidural recording an SD with reversed polarity? (Pages 17 and 25): Judging from Figure 4b, the first three SDs most likely start with a small initial negative DC shift, followed by the positive shift. This positive shift is clearly SD related, but it should not be considered an "inverted SD" and should not be used to calculate the duration of the depolarization. How likely are the bone screws to be the source of this positive shift if they do not affect all recording channels simultaneously?
What is the proportion of events that the authors consider to be "inverted SD" and how often can a negative shift of lesser amplitude be identified before the positive one?

2. SD continuum (Pages 7 and 29):

All SDs, regardless of the mode of induction, should be treated as a continuum (ref 12 and Dreier and Reiffurth 2015), and therefore the distinction between KCl induced SDs and SDs occurring during stroke evolution is not necessary. The features of SDs are highly dependent on the state of the tissue through which they propagate, so that SDs originating from the penumbra may be indistinguishable from SDs induced by the application of K⁺ (or by electrical stimulation, or by pinprick) when they reach completely healthy tissue. On the other hand, SDs induced in the healthy tissue and propagating into the penumbra will have features similar to the SDs generated in the penumbra itself.

Minor remarks:

After changing the term peri-infarct depolarization to spreading depolarization, this should also be changed in the title. The same applies to page 28 l 657

The term "invasive monitoring" (page 4, l 104) may be more appropriate than "aggressive monitoring", although this requires some qualitative grading. Subdural electrodes are obviously more invasive than epidural electrodes, but inherently less invasive than intraparenchymal electrodes.

References:

J. P. Dreier and C. Reiffurth, "The Stroke-Migraine Depolarization Continuum," *Neuron*, vol. 86, no. 4, pp. 902–922, May 2015, doi: 10.1016/j.neuron.2015.04.004.

Reviewer #3 (Remarks to the Author):

The authors have substantially revised the manuscript. However, my main concern (point #1) on statistical analysis remains. Each statement should be supported by quantifications, statistical descriptions and comparisons, ideally by additional figure panels. Below are few example points to ameliorate, but this applies to all statements in the results

Neural signals in all three frequency bands are larger amplitude than, and thus can be distinguished from system noise: add analysis, group data with indication of number of events and animals

Fig5c, Heatmap of RMS voltage before the SDs in the frequency range 0.5-200 Hz, revealing the border of the expanding penumbra: the difference between sites should be assessed through statistical tests, p-values should be presented. Ideally, comparisons also should be made with controls (before MCAO); if these are not available, with some "average" value from healthy tissue/ Group data should be provided as well

Same for the figure 7

Reviewer #1 (Remarks to the Author):

The authors have supplied thoughtful responses to all of the reviews, and added additional analyses and supplemental data where useful.

This added content has significantly improved the utility of the work for future investigations.

I have satisfied with their additions and corrections, and have no further comments.

Reviewer #2 (Remarks to the Author):

I would like to acknowledge the substantial improvements made in the current version of the manuscript. Nevertheless, a few issues still require careful consideration.

1. Is a positive DC shift in an epidural recording an SD with reversed polarity? (Pages 17 and 25):

Judging from Figure 4b, the first three SDs most likely start with a small initial negative DC shift, followed by the positive shift. This positive shift is clearly SD related, but it should not be considered an “inverted SD” and should not be used to calculate the duration of the depolarization.

How likely are the bone screws to be the source of this positive shift if they do not affect all recording channels simultaneously?

What is the proportion of events that the authors consider to be “inverted SD” and how often can a negative shift of lesser amplitude be identified before the positive one?

We have edited the manuscript for clarity, referring to “complex SD waveform” rather than “inverted SD” (page 18 of revised manuscript). We intended to say that in some cases, SDs are picked up by our array as having both a negative and positive components, with the positive waveform being more pronounced (as opposed to only a negative phase when detected by “true (glass)” DC intraparenchymal electrodes) – and **not** that two versions of SDs (inverted and non-inverted) exist. We have edited the manuscript to remove this confusion.

We believe the Reviewer is referring to Figure 5b (rather than Figure 4b), since pages 17 and 18 contain discussion of Figure 5b. The raw data from our arrays does indeed show SD events with positive components following smaller negative aspects, as previous groups have also documented (Kang et al. 2013). High-pass filtering the raw data results in a more pronounced positive phase (as shown in Figure 5b). We acknowledge that it is challenging to determine the duration of SDs that have both positive and negative components, which is why we emphasize the use of unfiltered data when calculating SD duration (pages 18-19 of revised manuscript). We also confirm that we do not use the positive waveform when calculating SD duration. To calculate SD duration, we first calculate amplitude, from the onset of the negative component and extending to the maximum negative deflection point. Then, we measure duration at half maximal amplitude.

While bone screws may pick up some cortical activity and contaminate the ground as result, we believe this would affect all channels simultaneously, depending on the underlying cortical activity. The five bone screws are placed multiple millimeters apart (with the two farthest screws being approximately 10 mm apart). Since SDs travel 3-6 mm/min, it is possible that during a long SD event, the SD travels from the array to include the site of at least one of the bone screws. In this case, the array would pick up near-zero voltage values (assuming the SD is done traveling across the array), but the bone screw itself would pick up a negative aspect (assuming the SD is passing underneath the screw). Recording from the array with reference to the bone screw would thus result in a positive waveform due to subtraction. While this is possible, we believe this effect is greatly minimized by having 5 bone screws spread across the cortex and averaging across all screws, up to 10 mm apart from each other). This scenario might explain why we detect SDs with negative components followed by positive aspects. However, because other groups have also seen SDs with both negative and positive waveform components in epidural recordings (Kang et al. 2013), we believe it has more to do with characteristics of epidural recordings in general (ie, distance from the source, metal electrodes and grounds, etc) than with anything specific about our experimental setup. While we can't definitively conclude this from our data, we believe the source of positive components is more likely to be a result of the nature of the epidural (versus parenchymal) recordings, as opposed to simply a contaminated ground setup. We have edited the manuscript to include this discussion.

2. SD continuum (Pages 7 and 29):

All SDs, regardless of the mode of induction, should be treated as a continuum (ref 12 and Dreier and Reiffurth 2015), and therefore the distinction between KCl induced SDs and SDs occurring during stroke evolution is not necessary. The features of SDs are highly dependent on the state of the tissue through which they propagate, so that SDs originating from the penumbra may be indistinguishable from SDs induced by the application of K⁺ (or by electrical stimulation, or by pinprick) when they reach completely healthy tissue. On the other hand, SDs induced in the healthy tissue and propagating into the penumbra will have features similar to the SDs generated in the penumbra itself.

We have edited the manuscript to emphasize the dependence of SD morphology on the state of the tissue and distance from the "source" (as opposed to mode of induction).

Page 8 of revised manuscript:

"All SDs occur as a continuum, regardless of the mode of induction [1, 2]. The features of SDs (i.e. morphology) are dependent on multiple factors, including the state of the tissue (healthy or not) they are propagating through, the site of recording with respect to the cortex (i.e., parenchymal vs superficial to the cortex), the nature of the electrodes and associated grounds (i.e., metal vs ionic, balanced vs unbalanced), rather than simply on the mode of induction. Therefore, any distinction between KCl and MCAO induced SDs was not necessary when evaluating the performance of the array during acute experiments."

Page 31 of revised manuscript:

"The variable nature of SD waveform, morphology, and spread during ischemia (where SDs can originate in and travel through tissue of variable health) complicates their reliable recognition and detection compared to SDs in a completely healthy brain, which are more stereotyped and feature a classical and

often uniform negative shape (when recorded intracortically, in reference to an indifferent ground.”

1. Hartings, J.A., et al., *The continuum of spreading depolarizations in acute cortical lesion development: Examining Leao's legacy*. *J Cereb Blood Flow Metab*, 2017. **37**(5): p. 1571-1594.
2. Dreier, J.P. and C. Reiffurth, *The stroke-migraine depolarization continuum*. *Neuron*, 2015. **86**(4): p. 902-922.

Minor remarks:

After changing the term peri-infarct depolarization to spreading depolarization, this should also be changed in the title. The same applies to page 28 | 657

We changed the title from

“High-density cortical μ ECoG arrays concurrently track peri-infarct depolarizations and long-term evolution of stroke in awake rats”

To

“High-density cortical μ ECoG arrays concurrently track spreading depolarizations and long-term evolution of stroke in awake rats”

We have also changed “peri-infarct spreading depolarizations” to “spreading depolarizations” on page 28, line 657 (old manuscript). In the revised manuscript, this is on page 30 line 728.

The term "invasive monitoring" (page 4, l 104) may be more appropriate than "aggressive monitoring", although this requires some qualitative grading. Subdural electrodes are obviously more invasive than epidural electrodes, but inherently less invasive than intraparenchymal electrodes.

We have edited this sentence to say “invasive monitoring” instead of “aggressive monitoring.” In the revised manuscript, this is on page 4 line 107.

References:

J. P. Dreier and C. Reiffurth, “The Stroke-Migraine Depolarization Continuum,” *Neuron*, vol. 86, no. 4, pp. 902–922, May 2015, doi: 10.1016/j.neuron.2015.04.004.

Reviewer #3 (Remarks to the Author):

The authors have substantially revised the manuscript. However, my main concern (point #1) on statistical analysis remains. Each statement should be supported by quantifications, statistical descriptions and comparisons, ideally by additional figure panels. Below are few example points to ameliorate, but this applies to all statements in the results.

Neural signals in all three frequency bands are larger amplitude than, and thus can be distinguished from system noise: add analysis, group data with indication of number of events and animals

Fig5c, Heatmap of RMS voltage before the SDs in the frequency range 0.5-200 Hz, revealing the border of the expanding penumbra: the difference between sites should be assessed through statistical tests, p-values should be presented. Ideally, comparisons also should be made with controls (before MCAO); if these are not available, with some "average" value from healthy tissue/ Group data should be provided as well

Same for the figure 7

We have added statistical analyses for Figures 2, 3, 5, 7, and 8 to accompany all statements in the results. We have described our statistical approach in the Methods section. We have reported the results of our statistical analysis in the Results section as well as in the Figures and figure captions.

For transparency, we have provided a detailed and comprehensive overview of all statistical analyses in the Supplemental Material.

All the changes have been highlighted in yellow in the revised manuscript.

REVIEWERS' COMMENTS:

Reviewer #2 (Remarks to the Author):

In the current manuscript version, all points of criticism have been thoughtfully addressed by the authors resulting in a successful revision. Therefore I see no need for further changes and have no further comments on this well-written manuscript.

Reviewer #3 (Remarks to the Author):

I have no further comments